# Solution-processable, soft, self-adhesive, and conductive polymer composites for soft electronics

Peng Tan[1,2], Haifei Wang[1,2], Furui Xiao[1], Xi Lu[1], Wenhui Shang[1], Xiaobo Deng[1], Huafeng Song[1], Ziyao Xu[1], Junfeng Cao[1], Tiansheng Gan[1], Ben Wang [1] & Xuechang Zhou [1✉]

Soft electronics are rising electronic technologies towards applications spanning from healthcare monitoring to medical implants. However, poor adhesion strength and significant mechanical mismatches inevitably cause the interface failure of devices. Herein we report a self-adhesive conductive polymer that possesses low modulus (56.1-401.9 kPa), high stretchability (700%), high interfacial adhesion (lap-shear strength >1.2 MPa), and high conductivity (1-37 S/cm). The self-adhesive conductive polymer is fabricated by doping the poly(3,4-ethylenedioxythiophene): poly(styrene sulfonate) composite with a supramolecular solvent (β-cyclodextrin and citric acid). We demonstrated the solution process-based fabrication of self-adhesive conductive polymer-based electrodes for various soft devices, including alternating current electroluminescent devices, electromyography monitoring, and an integrated system for the visualization of electromyography signals during muscle training with an array of alternating current electroluminescent devices. The self-adhesive conductive polymer-based electronics show promising features to further develop wearable and comfortable bioelectronic devices with the physiological electric signals of the human body readable and displayable during daily activities.

[1] College of Chemistry and Environmental Engineering, Shenzhen University, Shenzhen, P. R. China. [2] These authors contributed equally: P. Tan, H. F. Wang. ✉email: xczhou@szu.edu.cn

Soft electronics with abilities to withstand various mechanical deformations are essential in the fields spanning from wearable healthcare monitoring to medical implants[1–4]. Reliability of such devices particularly relies on electrodes/interconnects that bridging different active components as well as biological tissues, which are expected to form conformal attachment on irregular and non-flat surfaces and maintain durable stability upon deformations[5–7]. To date, various soft materials and flexible structures have been developed to construct reliable and sensitive electrodes for purpose of monitoring physiological signals in real-time fashion[8,9], including hydrogels[10], carbon-based composites[11], liquid metal composites[12], metal composites[13,14] and conducting polymers[15]. However, biological tissues, e.g., skins and muscles, are typically soft (with a mechanical modulus of 60–850 kPa), and the contact areas interfacing with electrodes are always irregular and even dynamic[16]. Such significant mechanical mismatch and weak interfacial adhesion strength may increase the noise and decrease the sensitivity of physiological signals, or even cause the failure of devices. Reliable interfacial connections between electrodes and biological tissues are of increasing importance for soft electronics[17].

Poly(3,4-ethylenedioxythiophene):poly(styrene sulfonate) (PEDOT:PSS) is a commercially available conductive polymer composite with good solution processability, tunable electrical property, and biocompatibility[18,19]. As a proper candidate for bioelectronics, PEDOT:PSS films process high rigidity (young's modulus > 500 MPa), low stretchability (about 5%), and poor interface adhesion strength, which may induce uncomfortability and even immune response during their bio-applications[20,21]. To reduce the modulus and enhance the stretchability, various methods have been developed, including small molecular doping, polymer blending, and hydrogel network construction[22–25]. However, the rigidity and modulus of as-doped PEDOT:PSS-based composites are still higher than that of human tissues. Moreover, undesirable plastic deformation is another issue that may cause incomplete recovery after deformation. Furthermore, the potential leakage of dopants (ionic liquids) may depress the performance of materials and cause a critical health issue to surrounding tissues. Hydrogels are commonly applied to fabricate ultrasoft PEDOT:PSS-based composites[17]. The limited conductivity (<10 S cm$^{-1}$) of PEDOT: PSS-based hydrogels is still a critical issue as a trade-off exists between softness and conductivity[26–28]. Apart from the mechanical mismatch, interface mechanical adhesion also needs to be improved. Approaches like electrogelation, electrografting, and the introduction of an adhesion layer have been reported to enhance interface adhesion stability of PEDOT[29–31]. However, these approaches usually require complicated interface modification or specific electrochemical deposition process and the adhesion of the PEDOT layer to the substrate is disposable and non-recyclable.

Herein, we developed a self-adhesive conductive polymer (SACP) composite for soft electronics by doping the rigid and nonstick PEDOT:PSS composites with a biocompatible supramolecular solvent (SMS). SACPs are prepared by the mixing and drying of a mixture containing SMS (i.e., citric acid and cyclodextrin), elastic polymer network (poly (vinyl alcohol) (PVA) covalently crosslinked by glutaraldehyde (GA)), and conductive polymer (PEDOT:PSS) (Fig. 1a). The SACP composites possess exceptional mechanical flexibility (low modulus, low residual strain, and high stretchability), high conductivity, and strong interface adhesion strength. SACP films and patterns can be fabricated by solution processing (i.e., drop-casting, spin-coating, microfluid molding, and even transfer printing) on diverse substrates, such as metals, glass, polytetrafluoroethylene (PTFE), polyimide (PI), polydimethylsiloxane (PDMS), etc. The SACP interface shows high adhesion strength (e.g., lap-shear strength up to 1.2 MPa on PI substrate), low interface electrical impedance, and good reusability (over 100 repetitions). We demonstrated the application of SACPs in a series of soft electronic devices, including alternating current electroluminescent (ACEL) devices (<150 μm in thickness), adhesive bioelectrodes for electromyography (EMG) monitoring, and an integrated system combing EMG sensors with an array of ACEL devices for the visualization detection of EMG signals during muscle training.

## Results

**Design and fabrication of SACPs for soft electrical interfaces.** Electrical interfaces for soft electronics require both high bonding stability and a low interface electrical impedance to maintain a good electrical signal transmission. The bonding performance relies on the adhesion strength and modulus (Fig. S1), which may cause delamination, wrinkles, or slippage, resulting in the collapse of interfacial electrical connections (Fig. S1, i–iii)[32,33]. Hence, a soft electrical interface of low modulus, strong adhesion, and high electrical conductivity is capable to address this issue (Fig. S1, iv).

SACP is one of the suitable candidates for soft electrical interfaces, owing to the low modulus, strong adhesion, and high conductivity. SACPs consist of three components, i.e., SMS (citric acid and cyclodextrin with a molar ratio of 10:1), elastic polymer networks (chemically crosslinked PVA networks with GA), and conductive polymers (PEDOT:PSS) (Fig. 1a). Briefly, citric acid, cyclodextrin, PVA, and GA were successively added to the aqueous solution of PEDOT: PSS. A homogeneous mixture was obtained and served as a "coffee-ring" free ink (Fig. 1b). SACPs showed various applications in soft electronics owing to the following aspects (Fig. 1c–d). First, supramolecular interactions occur from hydrogen bond interactions of a large number of carboxyl groups and hydroxyl groups in SMS[34]. Meanwhile, SMS can form hydrogen bonds and electrostatic interactions with PSSH units and positively charged PEDOT, respectively (Fig. 1a)[35]. Such interactions significantly inhibit the aggregation of PEDOT chains (Figs. S2 and S3), thus increasing the degree of freedom of the polymer chains[36], and thereby improving their mechanical flexibility. Second, PVA polymer networks, which are formed due to the selectivity of chemical crosslinking of GA with PVA and β-CD (Figs. S2a–b and S4), are constructed in PEDOT: PSS composites to obtain large yet reversible stretchability. Notably, the addition of dopants may inevitably cause plastic deformation[23]. Weak internal networks may cause greater plastic deformation upon stretching[28]. PVA networks solve the problem of irreversible deformation[37], resulting in good recovery of SACPs (Fig. S5). Third, owing to the existence of abundant hydroxyl functional groups and charged molecules, SACPs exhibit strong interface adhesion on substrates due to the synergistic effects of several weak interactions at the interface[38]. Fourth, the components are all soluble in water and capable of forming a homogeneous, stable, and viscosity-tunable aqueous ink (with viscosity on the order of $10^3$–$10^4$ mPa s, Fig. S6), rendering it particularly suitable for solution processing towards the scalable fabrication of flexible devices[39].

**Mechanical and electrical performance of SACPs.** Previous studies indicate that PEDOT:PSS composites possess a compromise between low modulus and high conductivity. Here we demonstrated low modulus and high conductivity can be simultaneously obtained in SACPs.

We first investigated the doping effect of SMS on the mechanical properties of SACPs. Stress-strain curves of the SACP films with different mass ratios of PEDOT:PSS were measured by tensile testing. The mechanical property of SACP

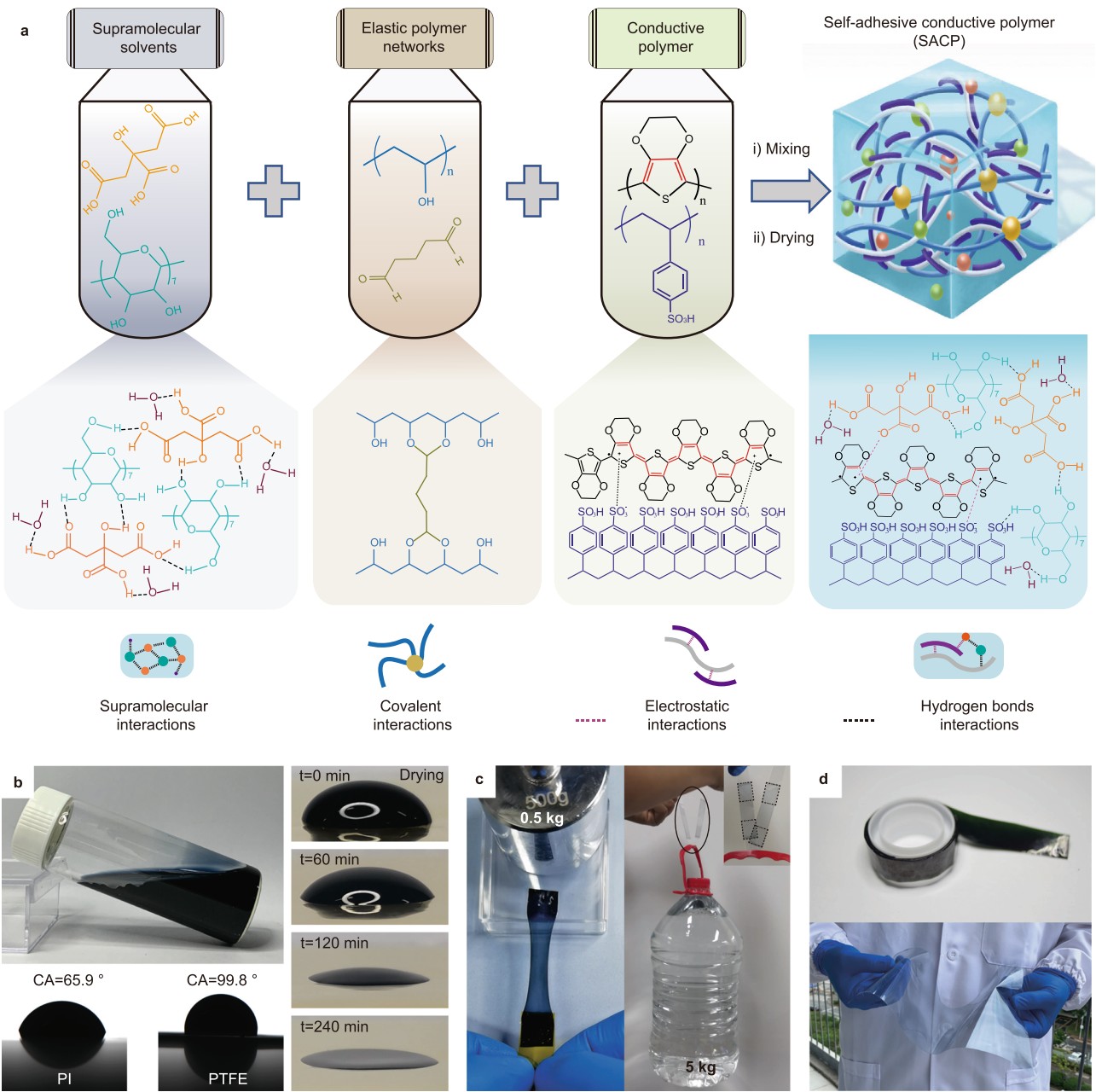

**Fig. 1 Fabrication of self-adhesive conductive polymers (SACPs) with supramolecular solvent-doped PEDOT: PSS composites and demonstration of their key features. a** Schematic illustration of the fabrication, molecular structures, and interactions of the SACPs, including chemical structures of PEDOT, PSS, citric acid (H3Cit), poly (vinyl alcohol) (PVA), glutaraldehyde (GA), and β-cyclodextrin (β-CD), and the involved interactions enabled by covalently crosslinked elastic networks, electrostatic interaction, and supramolecular solvents. **b** Images of the aqueous SACP ink, droplets on PI (CA 65.9º) and PTFE (CA 99.8º) substrates, and the drying process of a droplet on PI substrate. No coffee ring effect is observed during the drying process. **c** Soft and strong adhesion by SACPs: images showing the pulling of a plastic box loading with a 0.5 kg weight (left) and the lifting of a 5 kg water bottle (right) via the SACP-based adhesives. **d** Large-scale application of the SACPs: images of a roll of SACP-coated PET tapes (top) and an A4 SACP-coated PET sheet (20 × 25 cm) (bottom).

with a 36.3% mass ratio of PEDOT:PSS is hard with fracture strain of 154%, young's modulus of 5.9 MPa, and fracture stress of 14.8 MPa. While, the mechanical property of SACP with a 3.6% mass ratio of PEDOT:PSS is soft with fracture strain of 736%, Young's modulus of 401.9 kPa, and fracture stress of 1.2 MPa. We observe a significant decrease in Young's modulus and an increase in stretchability after doping. As the mass ratio of PEDOT: PSS decreases from 3.6% to 0.9%, the elastic modulus and fracture stress of SACPs gradually decrease to 56.1 ± 13 kPa and 593.2 ± 178.2 kPa. While the fracture strain slightly increases

up to 700% at low PEDOT:PSS content (mass ratio <3.6%) (Fig. 2a, b and Fig. S7a). The modulus of SACP can be tuned compatibly to that of skin tissue[16]. SMS doping provides an effective strategy to soften the PEDOT: PSS composites in a wide range.

Not limited to SMS doping, the mechanical properties of SACPs can also be further improved with PVA networks as well as the water drying process. Residual strain is one key parameter for soft materials when they need to be stretched in practical applications. By introducing a PVA elastic network, we lower the

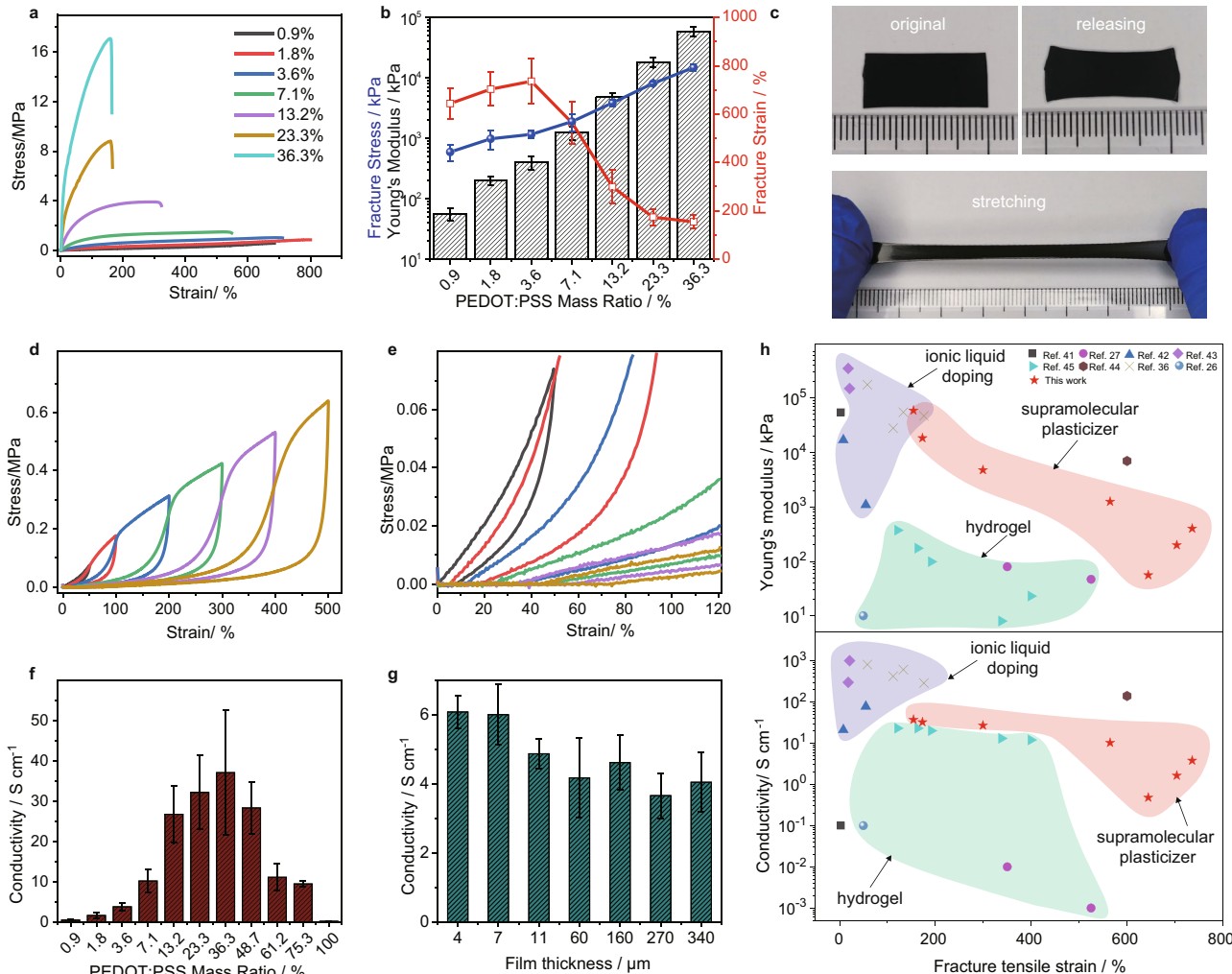

**Fig. 2 Mechanical and electrical performance of SACP films. a** Stress-strain curves. **b** Young's modulus, fracture stress, and fracture strain of the free-standing films with different PEDOT:PSS loadings. **c** Images of the films (with PEDOT:PSS mass ratio of 3.6%) presenting little residual strain even after being stretched to 400% of strain. **d** Stress-strain curves of the films (with PEDOT:PSS mass ratio of 3.6%) under cyclic loading-unloading tests. **e** Magnifying stress-strain curves between 0–120% for checking residual strain after stretching. **f** Conductivity of the films with different PEDOT:PSS loadings. **g** Conductivity of the films (with PEDOT:PSS mass ratio of 3.6%) with different thicknesses. **h** Relationships of Young's modulus and conductivity versus fracture strain of a variety of PEDOT:PSS composites reported in references and this work. The error bars show the standard deviations from at least three tests.

strain residual of PEDOT:PSS composites. SACPs with a 3.6% mass ratio of PEDOT:PSS show a large range of reversible stretchability (400%) (Fig. 2c), and the residual strain is <50% even at a large tensile strain (Fig. 2d and e). Not only strain residual, but we also observed the improvement of elastic resilience as the increase of concentration of the GA crosslinker (Fig. S7b and Supplementary Video 1). In addition, the mechanical performance is significantly changed as water evaporation time varies (Fig. S8), due to the impacts of water molecules on the hydrogen bond interactions in SMS.

Along with the tuning of mechanical properties of SACPs, the electrical conductivity also shows remarkable controllability ranging from 1 to 37 S/cm. We investigated the conductivity of the SACPs at a different mass ratio of PEDOT:PSS (Fig. 2f). Interestingly, we observed an increase in the conductivity at the small mass ratio and a gradual decrease at the large mass ratio, with a maximum conductivity of 37 S/cm at a 36.3% mass ratio. Given that the SMS and PVA networks are insulating, the conductivity of SACPs compliance percolation theory[22,40]. Therefore, there is a trade-off between mechanical flexibility

and conductivity. Besides, the as-made composites can be prepared as conductive films. Interestingly, the as-made SACP films with different thicknesses (4–340 μm) by spin-coating (for thin-film) and casting (for thick-film) methods showed good uniformity in electrical conductivity (Fig. 2g). According to the dynamic electrical resistance variations test (Fig. S9), the relative electrical resistance changes <1 at a strain of 100%, and the variation values remained constant under repeated strain-releasing cycles. This result indicates that the SACP films present good stability of resistance at a tensile strain within 100%.

Figure 2h shows a comparison of Young's modulus and conductivity of PEDOT:PSS and their composites. Intrinsic PEDOT:PSS has a large young's modulus (>500 MPa), low stretchability (<5%), and low conductivity (<1 S/cm) (Fig. 2h, black square)[41]. And it is difficult to reconcile mechanical and electrical properties. For example, ionic liquid-doped PEDOT: PSS composites have high conductivity (>100 S/cm mostly) (Fig. 2h, purple region)[36,42,43]. However, such composites exhibit a high young's modulus (>10 MPa), poor stretchability (fracture tensile strain <200%), and serious plastic deformation. Recently,

to produce elastic PEDOT:PSS composites with low young's modulus and high conductivity, ionic liquids plasticizers were used to adjust the microstructure of PEDOT complexes. For example, polyurethane and ionic liquid-doped PEDOT:PSS composites have been achieved at a modulus of 7 MPa and electrical conductivity of 140 S/cm[44]. In contrast, PEDOT: PSS hydrogel was developed with some features of low modulus (<1 MPa) and large ductility (200–600%) (Fig. 2h, green region)[26,27,45]. Although hydrogels are of great significance in the application of bioelectronics, the conductivity of hydrogel materials is much poorer (0.01–23 S/cm) as compared to their counterparts. What is more, the strain residual is typically larger than 100% for hydrogels and there is also serious plastic deformation. The present SACP exhibits relatively good electrical properties (up to 37 S/cm) and much superior mechanical properties, i.e., lower modulus (up to 56 kPa), low strain residual (<50% at 500% tensile strain), smaller plastic deformation, and larger stretchability (up to 700%), rendering it one promising candidate for soft electronics. Taking into account the compromise in mechanical flexibility, conductivity, and interface adhesion (in following discussions), SACPs with PEDOT:PSS mass ratio of 3.6% presented suitable mechanical property (modulus of 401.9 kPa) and conductivity (3.79 S/cm) meet the requirements of bioelectrode.

**Interface adhesion performance**. Traditional PEDOT:PSS films will break apart during repeated bending cycles, and the poor interface adhesion makes them easy to peel off from the substrate (Fig. 3a). The present SACP film exhibits high adhesion performance on PI and elastic Ecoflex substrates (Fig. 3b, c and Supplementary Video 2). The adhesion force of the SACP film on the substrate can tolerate a variety of load conditions (hanging, pulling, and lifting) (Fig. 1c and Fig. S10), and even applicable to biological tissues such as the liver surface of pigs (Fig. 3d).

We first investigated the adhesion strength of SACP by applying a 180-degree peeling test (Fig. 3e). The results show that the adhesion depends on the concentration of PEDOT: PSS in SACP films. The adhesion force increases as the PEDOT: PSS content decreases, and the interfacial adhesive force is about 400 N/m for the PEDOT:PSS content at 0.9% (Fig. 3f). Importantly, the self-adhesive film shows fast bonding ability (30 s) (Fig. S11), owing to multiple interactions such as hydrogen bonds, ionic interaction, and Van der Waals' interactions[46]. Notably, the SMS of citric acid and cyclodextrin has good interfacial adhesion (several MPa)[34]. More importantly, the SACP film shows substantially high performance of adhesion on diverse substrates (>150 N/m for PI; >120 N/m for PEEK; >100 N/m for Al, Cu, and PET; >80 N/m for PTFE; >30 N/m for PDMS) (Fig. 3g). The differences in interfacial adhesion forces of various polymer substrates were mainly dependent on molecular polarization strength of polymer chains and variations in chemical structures[10,47]. The high adhesion strength on PI or PEEK substrate can be ascribed to the strong dipole interactions of –C=O groups of their polymer primary chains. Meanwhile, PE or PDMS substrate has low interface adhesion strength due to their weak dipole interactions of polymer primary chains. Besides, the interface roughness and interface energy of substrate are also important factors affecting adhesion strength. The interface adhesion strength is even better than that of commercial test tapes (Fig. S12). The strength of interface adhesion on the substrate is strong enough to overcome the bending stress of different substrates (TPU, PC, PI, and TPFE) (Fig. 3h and Supplementary Video 3). Besides, the SACP films show remarkably high stability in storage and repeated usage. We did not observe any obvious decay in adhesion strength by storing the

film (temperature at 30 °C, humidity of 40%) for 30 days (Fig. 3i). Although we observed the adhesion strength reduced by 25% for a second time of the adhesion on PI substrate, the adhesion performance remains stable in the subsequent 100 cycles (Fig. 3j). This feature of repeated adhesion performance may be originated from physical interaction on the interface, avoiding irreversible damage of delamination[47].

We further investigated the lap shear strength of SACP on diverse substrates. The lap shear test was conducted by depositing the SACP film (<50 μm in thickness) on diverse substrates (Fig. 3k). We observe that the shear adhesion force increases from 0.6 to ~1 MPa as the thickness of the film increases from 7 to 20 μm as shown in Fig. 3l, and finally reaches a stable adhesion strength (over 1.2 MPa) from 28 to 43 μm, which is larger than that of the PU adhesive layer[30]. Such a strong lap shear strength can also be visualized by the lifting up of a 5 kg weight by the overlapped film (Fig. 1c). Like the adhesion strength, the lap shear strength shows a similar trend relating to different substrates (Fig. 3m). However, lap shear adhesion strength is much greater than the peeling adhesion strength, indicating a significant adhesion anisotropy in the adhesion performance of the SACP thin film. The strong lap shear adhesion strength is of significance to improve the stability and durability of SACP thin-film devices.

**Solution processability**. SACP can be processed by various solution-processing techniques owing to its excellent stability, homogeneity, and "coffee ring" free patterning capability. Although the contact angle of the SACP solution shows a slight fluctuation on different substrates (Fig. S13), the fluid dynamics of the droplets and the resultant patterns on the substrate are consistent (Fig. S14). As a proof-of-concept, we demonstrated the fabrication of SACP films and patterns by various solution process techniques as well as their applications in transfer printing and flexible ACEL devices.

Figure 4a–c shows the solution-processing fabrication of SACP films and patterns, e.g., microfluid molding, drop-casting, and spin-coating. SACP films can be deposited in microchannels, leading to the formation of "U" shape films on the walls of microchannels (Fig. 4a and Fig. S15). The combination of drop-casting and laser patterning techniques enables the fabrication of SACP-patterned Ecoflex. As shown in Fig. 4b (i)–(iii), the as-made patterned film showed no delamination from Ecoflex substrate under repeated stretching, indicating high adhesion stability of SACPs on the soft dynamic surface. In addition, a transparent conductive film with a maximum area of over $8 \times 8 \, cm^2$ can be obtained by spin coating (Fig. 4c), which can serve as transparent electrodes in photoelectric devices[48]. We also demonstrated the fabrication of large-size ($13 \times 19 \, cm^2$) transparent yet adhesive thin film on PET substrate by using Meyer rod coating method, and the corresponding SACPs conductive film can be used as electrodes of a large-area ACEL device with an area of $9 \times 14 \, cm^2$ (Fig. S16). The transmittance of the SACP film is close to 70–95% at 550 nm (Fig. 4e). The as-made SACP film exhibits square resistance of about 1000 Ω/□ with the light transmittance at 90% (Fig. S17a) and good bending stability (Fig. S17 b, c). We further demonstrated the transfer printing of the SACP patterns onto PI from PTFE substrates according to their difference in adhesion (Fig. 4d and Supplementary Videos 4). We also demonstrated the fabrication of SACP films on diverse substrates (PDMS, PET, PI, TPFE, glass, and copper) by spin coating (Fig. 4f). The transparent conductive film on the PET substrate was not broken even pasted by commercial test tape several times and no obvious electrical resistance changes were observed (Fig. 4g and Fig. S17d). The self-adhesive conductive film can be cut and adhered face-to-face to form a mechanically

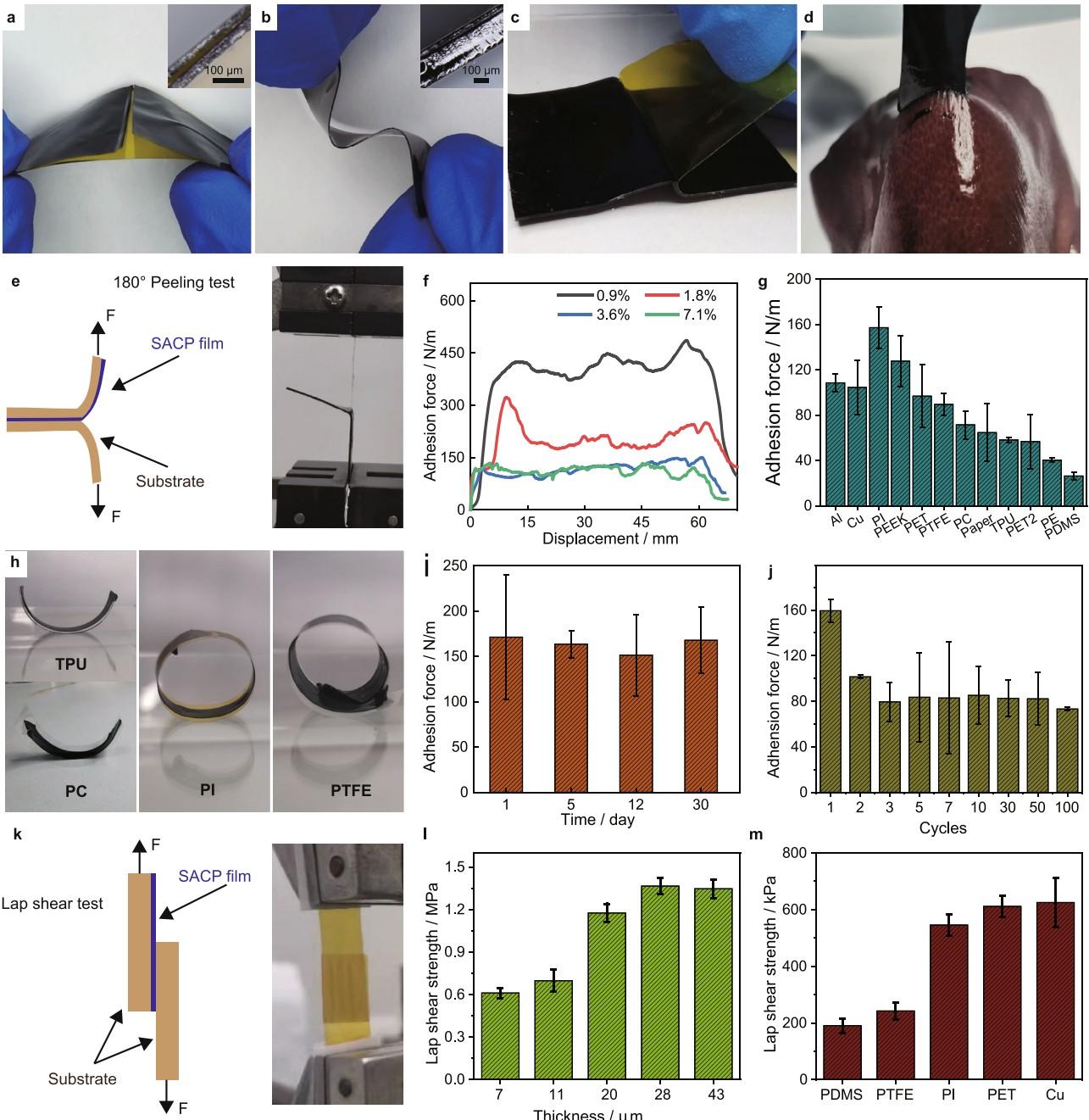

**Fig. 3 Interface adhesion performance of SACP films. a–b** Images of pure PEDOT:PSS (**a**) and SACP (**b**) films on PI substrate, insert: magnifications of adhesion interfaces. **c** Image of peeling off PI film from the adhesive film. **d** Image of adhesion between adhesion film and liver of a pig (the liver surface was cleaned by absorbent paper). **e** Illustration (left) and image (right) of 180° peeling tests of SACP films. **f** Measured peeling force per width of SACP films with different PEDOT:PSS loadings on PI substrates. **g** Summary of adhesion strength of SACP films on a wide variety of substrates. **h** Images of the bending state of substrates by releasing pre-stretched SACP film-adhered substrates. **i** Adhesion stability of SACP films after being stored in the constant temperature and humidity (temperature 30 °C, humidity 40%) for different days. **j** Adhesion strength of cyclic adhesion tests. **k** Schematic illustration (left) and image (right) of lap shear tests. **l** Lap shear strength of SACP films with different thicknesses. **m** Summary of lap shear strength of SACP films on different substrates. The error bars show the standard deviations from at least three tests.

durable and electrically conductive connection (Fig. 4h and Fig. S17e). Similar to the superiority of self-healing organic conducting film on electrical recovery after mechanical damage[49,50], this SACP film may also open a way to electrical recovery. Moreover, the SACP film exhibits a lower impedance as compared with the traditional PEDOT:PSS (Fig. S18).

We fabricated ACEL devices based on the SACP transparent thin-film electrodes by spin-coating. In the experiment, the bottom SACP electrode (7 μm), the light-emitting layer (ZnS:Cu particles and PDMS, 75 μm), the top SACP electrode (7 μm) and the PDMS encapsulation layer (35 μm) were spin-coated on the PET substrate (5 μm) layer by layer, and thus ACEL thin-film device was obtained (Fig. 4i). The thickness of ACEL film does not exceed 150 μm (Fig. 4j). ACEL devices have exceptionally high performance in flexibility and can withstand various deformations including bending, kneading, folding even multiple

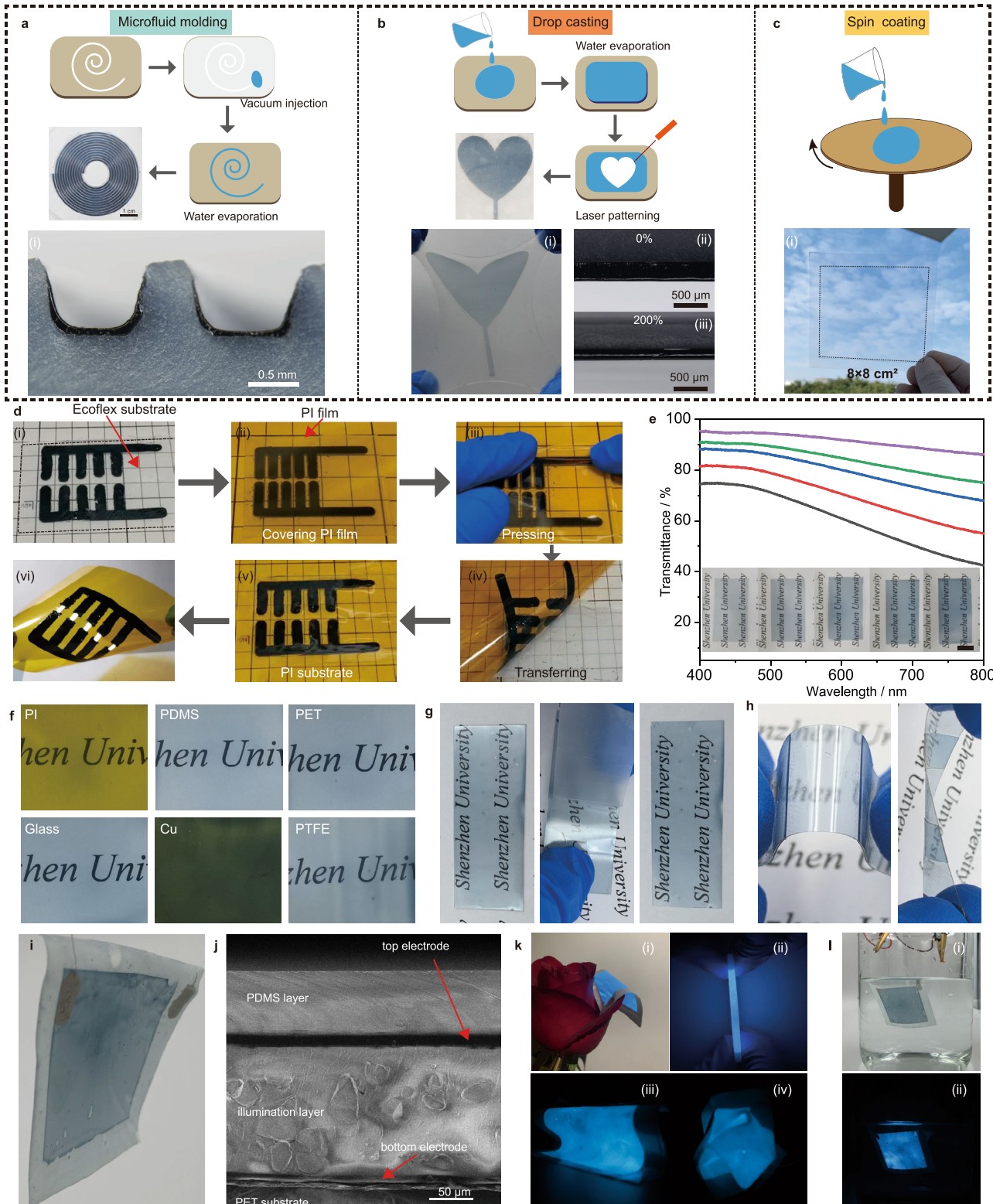

**Fig. 4 Solution processing and patterning of SACP patterns and thin films for flexible or even transparent electrodes. a** Microfluid molding. **b** Drop casting. **c** Spin coating. **d** Transfer-printing of SACP patterns for lighting up an LED lamp from Ecoflex to PI substrate enabled by the difference of adhesion strength. **e** UV–vis spectrum and images of different transmittance SACP films. **f** Images of SACP films on various substrates by spin-coating. **g** Images of SACP films under repeating peeling tests by commercial test tapes. **h** Images of adhered PET film under bending and twisting. **i** Image of alternating current electroluminescence (ACEL) devices by applying SACP thin film as transparent electrodes. **j** Scanning electron microscope (SEM) image of the cross section of the ACEL devices. **k** Images of the ACEL device (i) under folding (ii), bending (iii) and wrinkling (iv). **l** Images of the ACEL device immersed in water for 24 h under bright (i) and dark (ii) backgrounds.

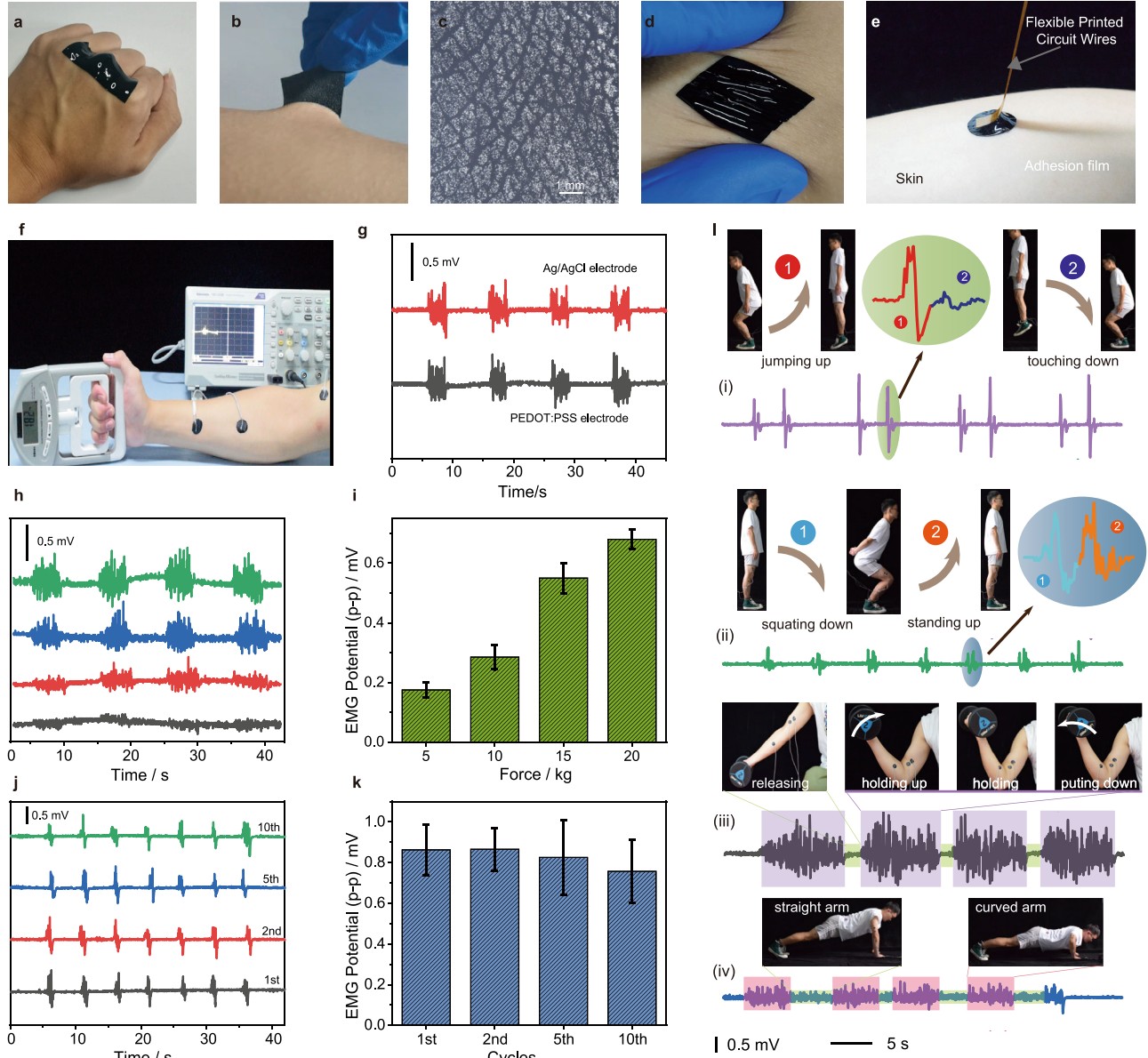

**Fig. 5 Skin adhesion and electromyography (EMG) monitoring of human exercises by using SACP electrodes. a–b** Adhesion of films on joints of fingers (**a**) and skin of arm (**b**). **c** Images of skin texture patterned on an SACP film after peeling off from the skin. **d** Images of SACP films adhered on the skin under distorting. **e** Interface connection between skin and flexible printed wires. **f** Image of an EMG test: SACP film electrodes were attached to the skin near the muscle, and gripping forces were displayed by a dynamometer. **g** Comparison of EMG signals by using commercial Ag/AgCl electrode (red line) and SACP film (black line). **h–i** EMG signals (**h**) and peak-to-peak value (**i**) under different gripping forces. **j–k** Stability of EMG signals (**j**) and corresponding peak to peak values (**k**) under repeated using test. The error bars show the standard deviations from at least three tests. **l** Detection of body exercises from EMG signal and the corresponding images (inset), jumping (i), squatting (ii), dumbbell lifting (iii), and push-up (iv).

folds (Fig. 4k, Fig. S19 a–c and Supplementary Videos 5). Not limited to the ultra-high flexibility, the ACEL device is highly stretchable when using an elastic substrate, and the emission intensity increased at low strain then decreased at high strain, because the electrical field is varied by the trade-off between the resistance and space of electrode under strain (Fig. S19 d–h). Such a thin and light structure allows the working ACEL device to flutter in the wind. In addition, the good interface adhesion performance renders it a good packaging, and as such the ACEL device can work stably even in water for 24 h (Fig. 4l).

**Physiological electrical potential monitoring.** The SACP film is an ideal candidate for bioelectronics owing to its low modulus,

high adhesion strength, and good conductivity[15,51]. The SACP film can conformally adhere to human skin (Fig. 5 a–c). The adhered film deforms synchronously when skin suffers from various mechanical deformations, i.e., stretching, compressing, and distorting (Fig. 5d). As a proof-of-concept, SACP films are applied as conductive interfaces between human skin and electromyograph (Fig. 5e and f). The biological EMG signals of a human hand are monitored at different gripping forces. The results reveal that the performance of the adhesive and conductive film electrode is comparable to a commercial Ag/AgCl electrode (Fig. 5g) and the other conductive materials in literatures[11,13,15]. By applying 5, 10, 15, and 20 kilograms of forces to the grip strength meter, the SACP electrode can monitor

the steady increase of the EMG signals (Fig. 5h and i). In addition, the EMG signals do not show significant attenuation even when the SACP electrodes had been used ten times (Fig. 5j and k).

We can also monitor various human-body activities by mounting the SACP electrodes on different muscle positions (Fig. 5l and Supplementary Videos 6). For example, by adhering it to leg muscles, we can monitor body motion by recording EMG signals of jumps and squats (Fig. 5l (i) and (ii)). We can record two characteristic EMG signals during a one-time jump or squat motion. For jumps, the EMG signals of jumping up were larger than that of touching down. While, the EMG signals of squatting down and standing up are almost the same, in the squat-up motion. Besides, we also recorded completely different EMG signal characteristics between dumbbell lifting and push-ups by adhering self-adhesive conductive films to arm muscles (Fig. 5l (iii) and (iv)). The EMG signals generated by continuous arm muscle contraction can be recorded.

**Integrations of EMG sensor and ACEL arrays**. The outputs of EMG signals can be further harvested to trigger the ACEL arrays[52]. The physical movement, especially the strength, can be directly visualized and monitored by ACEL devices. To validate this concept, we designed and fabricated an integrated system for the visualization of muscle training by ACEL. The system consists of two units, i.e., EMG monitoring and displayable ACEL array (Fig. 6a). The ACEL array is fabricated by spin coating (Fig. 6b). A set of EMG monitoring electrodes (Fig. 6c, (i)) attached to the forearm transmits the biopotential of the hand muscle to the EMG sensor (Fig. 6c, (ii)), and then the EMG signal is converted into the corresponding value through a signal analysis and control unit (Fig. 6c, (iii)). In accordance with EMG signals, the pixels of ACEL arrays are turned on/off depending on the threshold values, realizing the visualization detection of EMG signals of muscle motion (Fig. 6c, (iv)). As shown in Fig. 6d, by applying different levels of grip strength, the corresponding EMG signals are detected. After signal filtering analysis and processing, a set of control signals are generated to control the on/off of ACEL arrays. By setting appropriate switching thresholds, the ACEL arrays display according to the strength of gripping (Supplementary Videos 7). SACPs serve not only the bioelectrodes for harvesting the EMG signals but also the adhesive transparent thin-film electrodes for ACEL devices. This study demonstrates a concept of the direct visualization of the physiological electrical signals of the muscles during physical movement and paves the way to develop wearable bioelectronic devices that can visualize the activity strength of our daily life in a real-time fashion.

## Discussions

SACP is a type of soft, adhesive, and conductive polymer composite that is easily fabricated by doping the SMS into the rigid and nonstick PEDOT: PSS composite. Compared with the other kinds of PEDOT: PSS-based conductors, SACP simultaneously possesses low modulus, strong interface adhesion strength, and high conductivity in one material, rendering it an ideal candidate for soft electrical interfaces in soft electronics. By using SACP, we have fabricated self-adhesive conductive films/patterns via a variety of solution processing methods such as drop coating, spin coating, and microfluid molding, and even transfer printing of a pre-fabricated circuit between different substrates. The proof-of-concept demonstrations of the SACP, as soft and conformal bioelectronic devices, are carried out, including thin film ACEL devices and bioelectrodes for monitoring EMG signals of muscle during exercising. We finally verify the possibility to visualize muscle training by an SACP-based bioelectronic system that

integrates EMG sensors and ACEL flexible arrays. The SACP-based electronics show promising features to further develop wearable and comfortable bioelectronic devices with the physiological electric signals of the human body readable and displayable during daily activities.

## Methods

**Preparation of SACP films**. The SACP films were prepared by drying SACP precursor solution. The SACP precursor solution was prepared by mixing SMS solution, polymer solution, and conductive polymer solution. SMS solution comprised of β-cyclodextrin (β-CD, AR, Aladdin) and citric acid (H₃Cit, AR, Aladdin), which were mixed in water at the molar ratio of 1:10 under agitation[34]. Polymer solution contained poly (vinyl alcohol) (PVA 1799, DP of 1700, 98–99% hydrolyzed, Aladdin) aqueous solution as polymer network and glutaraldehyde (GA, 50% volume percentage, Macklin) as cross-linker. Finally, SACP precursor solutions were obtained by adding SMS solution and polymer solution into PEDOT: PSS aqueous solution (1.1–1.3% solid content, Clevios™ PH1000, Heraeus Electronic Materials) under vigorous agitation for 10 min. The pH value of SACPs precursor solution was about 3–4 due to the presence of citric acid. The crosslinking reaction of GA and PVA was triggered in presence of H₃Cit as an accelerator. The crosslinking density of the PVA network was controlled by adjusting the added volume of GA. The details of SACPs with various mass ratios of PEDOT:PSS were presented in Supplementary Table 1. The mass ration of PEDOT:PSS (X) in SACPs was calculated according to the following equation:

$$X = \frac{X_0 m_0}{X_0 m_0 + m_{SMS} + m_{PVA} + m_{GA}} \times 100\% \tag{1}$$

Where, $X_0$ is the weight percentage of PEDOT:PSS in original solution (≈1.3%); $m_0$ is the weight of PEDOT:PSS solution; $m_{SMS}$, $m_{PVA}$, and $m_{GA}$ is the weight of SMS, PVA, and GA, respectively.

For thick SACP films, we applied a casting method. Briefly, after degassing in a vacuum desiccator for 20 min, the SACP dispersion was poured into a polycarbonate (PC) mold and dried at 40 °C for 24 h. Thus, free-standing SACP films were obtained. The film thicknesses were controlled by changing the volume of the SACP precursor solution. For SACP thin films, we applied a spin-coating method. In a typical experiment, the substrates were cleaned by DI water and alcohol, respectively. Then surface hydrophilic treatment of substrates was conducted by plasma cleaning for 15 min. Thin SACP films were deposited on substrates by spin-coating at speed of 1000 rpm for 60 seconds and drying at 70 ºC for 5 min. By changing the conditions of spin-coating, SACP films with diverse thicknesses were obtained. For the microfluid molding method, a PDMS microchannel was obtained by solidifying PDMS precursor (10:1, Sylgard 184, Dow Corning) on a homemade silicon wafer template patterned with a photoresist. Then, the resultant PDMS microchannel was covered by a PET film with a hole punched on the end of the channel. The hole was sealed with an SACP precursor solution droplet (2–3 g) and then degassed in a vacuum desiccator. Afterward, the SACP precursor solution was injected into PDMS microchannel once the air pressure of the vacuum desiccator gradually returns to atmospheric pressure, due to the pressure difference between the inside and outside of the channel, as detailed in our previous work[53]. After removing the covered PET film and drying at 40 °C for 24 h, 3D deposited SACP films were obtained.

**Measurement of the electrical performance of SACPs**. Conductivity measurements of all samples were carried out using the 4-point probe method (Keithley 2400 digital multimeter, Keithley). A rectangle film of SACP composite (about 30–40 mm of length, 10 mm of width) was used to test sample electrical conductivity. The thickness of the sample film was obtained by an optical microscope. Silver paste (conductive silver paint, SPI) was cast on both ends of the sample as electrodes. Electrical conductivity was calculated from the following equation:

$$\sigma = \frac{1}{R} \frac{L}{w\,h} \tag{2}$$

where $\sigma$ is conductivity (S/cm) and $R$ is resistance (Ω). $L$, $w$, and $h$ denote for length, width, and thickness of the sample (cm), respectively. Conductivity values of each condition were obtained by averaging over a minimum of three to five measurements. Resistance vs. strain relationship scrutinized by concurrent measurement of mechanical tensile strain and resistance. The sample was clamped in a tensile stage (Tensile testing machine, CMT 6103, MTS) with silver paste cast on each end as the electrode and stretched at a constant tensile speed of 10 mm/min.

**Measurement of the mechanical and adhesion performance of SACPs**. Tensile strain tests of all samples were carried out by a mechanical testing machine with a 50 N load cell. The measurement was conducted at 25 °C with a relative humidity of 40–45%. The reported statistical data were obtained from at least three times tests of the same sample. Standard rectangular test samples (length ≈30–40 mm, width ≈10 mm) were clamped in the tensile stage and stretched at a tensile speed of 50 mm/min, if not stated otherwise. Cyclic loading-unloading tests were conducted by stretching the samples to a strain of 200% for 100 cycles. Step loading-unloading tests were performed at a multi-step strain from 50%, 100%, 200%, 300%, 400% and 500%, respectively.

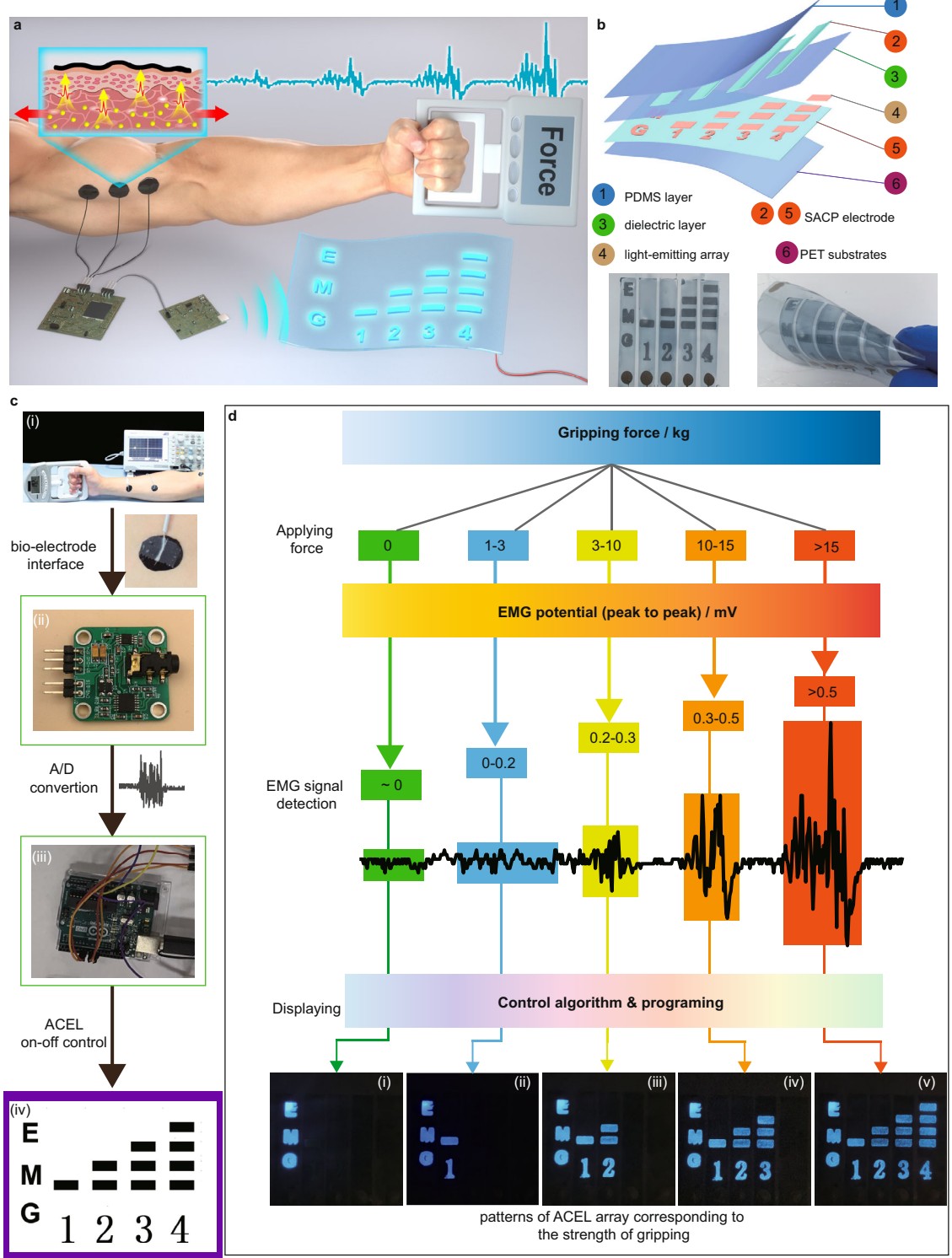

**Fig. 6 Demonstration of an integrated EMG visualization system with an array of ACEL devices. a** Schematic summary of the system with ACEL display controlled by EMG sensors. **b** ACEL display with patterned lighting arrays assembled by the transparent conductive film. **c** Four functional units in the system: (i) conductive film electrode for EMG signals transformation, (ii) commercial EMG sensors for data recording, (iii) commercial single-chip microcomputer for ACEL control, and (iv) flexible ACEL devices for EMG signals display. **d** Patterns of flexible ACEL display turning on and off based on the level of EMG signals.

Adhesion forces of free-standing SACP films adhering to various substrates were measured using a 180° peeling test with the mechanical testing machine. All substrates were treated using 3 M tape to remove surface contaminants. Then, free-standing SACP film was attached to the diverse substrates. The dimensions of the samples were 8-10 mm (width) and 60–80 mm (length). Loading-displacement curves were conducted with a tensile mold at a peeling test speed of 50 mm/min. To analyze the stability of adhesion force, various contact times (0.5, 1, 30, 60, and 120 min) of the substrate in contact with SACP film were conducted by measuring the adhesion force of 180° peeling test. Cyclic adhesion force tests were performed. The adhesion force was determined by averaging the peeling force ($F$) and then dividing the force by the width of the sample ($w$).

The interfacial shear strength was measured by a lap-shear test. The lap-shear test samples were prepared by spin-coating SACP film on the substrate. A film same to the substrate was covered on the upper surface of the SACP film with an overlap area of 10 mm (length) × 10 mm (width). The tap-shear tests were conducted at the tensile speed of 50 mm/min. The interfacial shear strength was determined by dividing the peak measured force by the overlap area.

**Preparation of ACEL devices**. The ACEL devices were prepared by spin-coating. A PET film (~10 μm in thickness) was used as the substrate, SACP films were used as top/bottom electrodes, and ZnS:Cu/PDMS was used as the emissive layer. The ZnS:Cu/PDMS composite was prepared by mixing ZnS: Cu microparticles (2 g, Shanghai KPT company) and PDMS (2 g, Sylgard 184, Dow Corning). The PET films were covered on glass and surface treated by a plasma cleaning machine (HARRICK Plasma PDC-002) with 400–500 mtorr in air for 15 min. The SACP dispersion was coated on PET film as the bottom electrode at a spin rate of 1000 rpm for 30 s, then dried at 70 °C for 5 min. The ZnS: Cu/PDMS composite was coated onto the SACP bottom electrode at a spin rate of 2000 rpm for 120 s. The ZnS:Cu/PDMS was thermally cured at 60 °C for 2 h. To improve deposition of the top electrode, ZnS:Cu/PDMS layer was surface treated by plasma cleaning for 15 min. The top electrode of the SACP layer was coated by a spin rate of 1000 rpm for 30 s and dried at 70 °C for 5 min. Finally, a pure PDMS package layer was coated on by a spin rate of 1500 rpm for 30 s. Silver wires were connected to the top and bottom SACP layer. The ACEL device was lighted by an input power purchased from the Shanghai KPT company.

**Monitoring of EMG signals**. To evaluate the performance of bioelectric communication of tissue-electrode interfaces, we recorded EMG signals by using the SACP films as bioelectrodes. Two conductive films were pasted on the forearm, and another conductive film was pasted on the upper arm, serving as three sets of electrodes for EMG signals monitoring. Then a certain force was applied to the grip meter by hand to stimulate the generation of electromyographic signals. For comparison between SACP films and commercial electrodes, the commercial Ag/AgCl electrode was also used. The EMG signals were recorded by a digital oscilloscope (TBS 1152B, Tektronix).

**The integrated system of EMG visualization detection by ACEL devices**. An integrated system of EMG sensor and ACEL array to realize the visualization detection of EMG signals. The system includes self-adhesive electrodes, EMG sensors, signal analysis controllers, and ACEL arrays. ACEL array was prepared by spin coating with a patterned template. The signal analysis controller is a single-chip microcomputer from Arduino, for processing EMG signals and controlling electronic switches. Each column of the ACEL array was controlled by electronic switches. The threshold of electronic switches was set into four levels in a single-chip microcomputer according to the EMG signals. The strength of EMG signal was determined by the single-chip microcomputer, and then was applied to control the ACEL array through electronic switches. The Arduino code for the integrated system of EMG visualization detection was provided in supplementary files.

**Morphological and chemical characterization**. Scanning electron microscopy (SEM) was executed on APREO S (Thermo scientific). SEM samples were prepared by drying under heat. Atomic force microscopy (AFM) was performed using a XE15 Litho AFM (Park Systems) in tapping mode with NCHR cantilever (spring constant of 42 N/m). Optical microscope images were obtained on an optical microscope (SMZ-18, Nikon) equipped with a CMOS camera (DS Ri-2). Optical images were taken with a digital camera (D7100, Nikon). The impedance spectra were taken on an LCR meter (Keithley) with the dual-electrode method in the ranges of $10^2$–$10^6$ Hz. UV–vis spectra were collected on a UV-2550 (Shimadzu) spectrophotometer. Apparent viscosity analysis was conducted by using a TA rheometer. The size distribution of PEDOT was measured using dynamic light scattering (DelsaMax CORE). The Surface element component was analyzed by X-ray photoelectron spectroscopy (Thermo Fisher Scientific) using Al Kα radiation. The Raman spectrum was recorded using microscopic laser confocal Raman spectrometer from Renishaw. The characterization of the chemical structure was measured by Fourier transform infrared spectrometer (IR Affinity-1, Shimadzu) and X-ray diffraction (D8ADVANCE, Bruker).

**Reporting summary**. Further information on research design is available in the Nature Research Reporting Summary linked to this article.

## Data availability

The data that support the findings of this study are provided as source data with this paper or included in the Supplementary Information. Further data are available from the corresponding authors upon request. Source data are provided with this paper.

## Code availability

Arduino code used for this study is shown the Supplementary information, which is available from the corresponding author on reasonable request.

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

## Acknowledgements

This work was supported by financial fundings awarded to Prof. X.C.Z. from the National Natural Science Foundation of China (21922303), Guangdong Basic and Applied Basic Research Foundation (2020B1515020045), and Shenzhen Municipality Science and Technology Planning Project (SGLH20180622151607182).

## Author contributions

P.T., H.F.W. and X.C.Z. conceived the concept and designed experiments. P.T. and H.F.W. conducted all the experiments and performed the mechanical characterization. P.T., H.F.W. and J.F.C. conducted the preparation of SACP. P.T., H.F.W. and X.L. performed electrical characterization. W.H.S., X.B.D., H.F.S., Z.Y.X. and T.S.G. assisted materials and devices characterization. P.T., H.F.W., F.R.X. and H.F.S. contributed to soft electronics fabrication and EMG measurement. H.F.W. and F.R.X. were involved in the soft electronics demonstration. T.S.G. and B.W. provide comments on the manuscript. H.F.W. and X.C.Z. were responsible for managing all aspects of this study. H.F.W. wrote the draft. B.W. and X.C.Z. revised the manuscript. All authors discussed the results and the manuscript.

## Competing interests

The authors declare no competing interests.
