## [Peer Review File · Nature Communications]

Solution-processable, soft, self-adhesive, and conductive polymer composites for soft electronicsREVIEWER COMMENTS

Reviewer #1 (Remarks to the Author):

This paper reports the fabrication of ultra-stretchable PEDOT:PSS composites and films and integration in a number of devices. There is a comprehensive mechanical characterization of free standing films. Overall this is a very comprehensive work, which is well written. I believe it could be published in NC after addressing the following points.

1. The authors report different compositions where the key components are clearly disclosed. However, the chemistry makes little sense to me. Why would cyclodextrin, which contains a -C-OH groups as PVA does, not react with GA? The reactivity of these OH groups are identical. This is a very important aspect to clarify since it is at the base of the major discovery of this work.
2. The authors also elaborate on the conductivity of these samples also at a very low loading of PEDOT:PSS based on percolation theory. This is obvious, otherwise the system would be an insulator. The question is why? Here more morphological and structural characterization of the films should be carried out. Equally important, are these solid solutions? If PEDOT:PSS aggregate, how, what are the domain size? Spectroscopic analysis can help as well IR/Raman.
3. Ref 44 should be discussed a bit more in the context of the work. Also, close analysis of the plot of Figure 2h seems not reconcile with the data reported in panel a and the conductivity of panel f.
4. The description of the SACP formulation preparation in lines 90-94 is confusing. Also, what is reported in Methods is not clear. Data are for many different samples fabricated in a way that is not reported. The experimental procedure for each samples has to be crystal clear. The same in some of the figures/text when referring to PEDOT:PSS (0.9% to 36.3% mass ration of PEDOT:PSS – versus WHAT?)
5. Skin is discussed in several places. What is the modulus range needed for the human skin. Please report.
6. Line 138, should be insulating not insulated.
7. Report viscosity range of formulations in the main text.
8. Experiments were carried out by placing the SACP films in contact with a human subject. This area of research is strongly regulated. One thing if a materials is inserted in a device and then the substrate is placed in contact with the skin. Here, I have concerns in carrying out experiments on a human without addressing safety concerns. For instance, what is the pH of these films? How long are they in contact with the human subject? Does this person understand the risk of this experiment?
9. The method section must be improved. Details for formulation preparation should be included. Also, for the mechanical tests, what is the RH? In general how many samples were measured for reporting statistical data?

Reviewer #2 (Remarks to the Author):

In this manuscript, the authors developed a self-adhesive conductive polymer (SACP) composite based on PEDOT:PSS, PVA, GA, and biocompatible supramolecular solvent. Furthermore, the authors investigated the mechanical and electrical properties of the composite film and demonstrated future practical applications such as electrophysiological sensing electrodes and advanced integrated systems based on the film. Although the concepts of this work, figures, and demonstration in this manuscript are interesting and exquisite, the reviewer cannot find enough strength in terms of scientific novelty and practical specialty. Especially, the highest softness, the highest adhesive, and the highest conductivity of the film show in different composite ratios, which represents that there is a trade-off. In addition, the authors did not even that optimize such properties. Moreover, the reviewer found quite many concerns from this manuscript. Therefore, the reviewer thinks this manuscript is not suitable for a high-ranked journal such as Nature Communications. More specific comments are listed below.

1. The authors mentioned that “PVA polymer networks are constructed in PEDOT: PSS composites to obtain large yet reversible stretchability.”. However, the reviewer believes that the authors need to

describe more experimental results such as morphologies to support the claim.

2. In Fig. 2, the author demonstrated the mechanical and electrical performance of SACP film based on only a 3.6 % mass ratio. The reason seems that the mass-ratio-based film shows the highest fracture stress. However, no description to clarify it in the manuscript.

3. In addition, it is helpful to understand the film properties comprehensively if the mechanical properties of 36.3% mass-ratio-based film are investigated because the film shows the best conductivity.

4. The SACP film cannot return to its original length after stretching ~400% in Fig.2c. Based on Supplementary Fig.6, it seems that the film was undergone plastic deformation. So, Fig2d needed to be obtained by different samples but no description in the manuscript.

5. Why the thinnest film shows the highest conductivity? Typically, the thicker film has better conductivity, especially such micrometer-thick range. However, the author needs to clarify it to improve the quality of this manuscript.

6. Label for the x-axis of Fig.2h should not be a strain but the fracture tensile strain.

7. The authors should clearly describe the liver surface of a pig when they demonstrate the applicability of the film adhesion onto the biological tissue since the wet or dry surface of the organ has a huge difference in terms of adhesion.

8. Fig.3g shows quite a different adhesion force of the film depending on substrate. However, the authors never mention and try to investigate the scientific reason. Such a lack of chemistry weakens the quality of the manuscript.

9. On page 11, the authors mentioned, "The as-made patterned film exhibits high stability under repeated stretching (Figure 4b)". However, the authors need to be careful of such description without investigating its conductivity change or microscopic view such as SEM image since it is possible to have lots of microcracks although it is not visible by bare eyes.

10. The image quality of supplementary Figure 16 needs to be improved. The reviewer cannot get any idea from the figure, although the authors mentioned it is an SACP film.

11. To be solid for the author's description, "The transparent conductive film on the PET substrate was not broken even pasted by commercial test tape several times (Figure 4g)." in page 11, conductivity and microstructure need to be compared before and after repeating peeling tests by commercial tapes.

12. The reviewer suggests a more systematic investigation of the ACEL performance, such as luminance under the different mechanical strain or deformation.

13. In the Methods, the authors need to clearly describe the microfluid molding method and transfer printing through schematic illustration or detailed description.

14. On page 16, the authors have to elucidate what kind of gas-based plasma is used for cleaning since the properties will differ depending on the gases.

15. The reviewer thinks it is better to provide Arduino code for the integrated system in the supplementary information.

Reviewer #3 (Remarks to the Author):

The work by Tan et al. is exciting and tackles one key challenge that is often overlooked in wearable organic electronics, i.e., the materials surface adhesion. In this manuscript, the authors demonstrated the possibility of obtaining a conducting material capable of acting as tape and withstanding high mechanical stresses. Overall, this manuscript is a perfect fit for Nature Communications and leaves little criticism. There are, however, a few issues that still need to be addressed before publication:

1) While the mechanical properties and devices are well characterized, I found the chemical characterization of this polymeric system lacking. The cartoon reported in Figure 1 provides a schematic overview of what could happen in this blend. However, there are no chemical data to support this claim.

2) How stable are these composites in a water and humid environment? The authors briefly mentioned stability in water of the ACEL device, where the top layers provide some packaging effect for the layer underneath. Yet, a proper characterization of the water stability is missing.

3) The conductivity values shown here are indeed good. Yet, as the authors correctly reported, there is a trade-off between high electrical conductivity and good mechanical properties. Therefore, a more in-depth analysis of what is more important for the selected applications is needed.

4) The state of the art is well discussed. Yet, some recent literature on self-healing conducting polymers (see for example, *Adv. Funct. Mater.* 2019, 29, 1905426 and *Adv. Funct. Mater.* 2020, 30, 2002853) is missing, and it is unclear how the current work relates to those. What are the self-healing properties of the conducting polymers under investigation?

Overall, this is an interesting manuscript, and the data reported here deserve publication in *Nature Communications*.

Response to Reviewer #1:

This paper reports the fabrication of ultra-stretchable PEDOT:PSS composites and films and integration in a number of devices. There is a comprehensive mechanical characterization of free-standing films. Overall this is a very comprehensive work, which is well written. I believe it could be published in NC after addressing the following points.

Response: Thanks for the positive comment and kind suggestions. The point-to-point responses are presented below.

1. The authors report different compositions where the key components are clearly disclosed. However, the chemistry makes little sense to me. Why would cyclodextrin, which contains a -C-OH groups as PVA does, not react with GA? The reactivity of these OH groups are identical. This is a very important aspect to clarify since it is at the base of the major discovery of this work.

Response: Thanks very much for the kind suggestion. Yes, we agree with the reviewer's opinion. The selectivity of the chemical crosslinking of GA with PVA and cyclodextrin is very important for understanding the formation of elastic polymer networks. As the reviewer stated, the reactivity of these -OH groups both from PVA and cyclodextrin are identical, while they have very different stereochemistry structures. And thus the reaction selectivity varies considerably. First, cyclodextrin is a typical class of oligosaccharides with chair conformation, while PVA is a linear chain structure. GA reacts simultaneously with two -OH groups to form O-C-O structure. For cyclodextrin, any two adjacent -OH groups are not in the same molecular plane because of the chair conformation of β -cyclodextrin (β -CD). The formation of O-C-O structure is unfavorable. While the large steric hindrance of cyclodextrin prevents glutaraldehyde from reacting with -OH groups from two cyclodextrin molecules. For PVA, -OH groups from the linear chain of PVA have favorable stereochemistry structure because of the free rotation of C-C chains. Therefore, the reaction selectivity of OH groups between PVA and β -cyclodextrin is different because of the difference in stereochemistry structure.

We also conducted experiments to validate the selectivity and the results are shown in Figure R-1 (also added as Figure S2a and 2b in the supplementary file). As shown in **Figure R-1**, only PVA polymer solution was gelled after the addition of GA. While β -CD with different stereochemistry structures of -OH groups can't be crosslinked by GA. Therefore, the chemically crosslinked polymer networks were formed from PVA chains in our work.

To make it clear, we added the following changes in the revised manuscript:

1) Page 6: "PVA polymer networks, which are formed due to the selectivity of chemical crosslinking of GA with PVA and β -CD (Figure S2a and 2b), are constructed in PEDOT: PSS composites to obtain large yet reversible stretchability."

2) Page 3 in the revised supplementary file:

"According to Supplementary Figure 2a and 2b, PVA polymer solution was gelled after the addition of GA. While β -CD can't be crosslinked by GA. This is because cyclodextrin is a typical class of oligosaccharides with chair conformation, while PVA is a linear chain structure. GA reacts simultaneously with two -OH groups to form O-C-O structure. Therefore, the chemically crosslinked polymer networks were formed from PVA chains."

Figure R1-1. The reaction selectivity of -OH groups in PVA (a) and β -CD (b) for the formation of the chemically cross-linked polymer gel.

2. The authors also elaborate on the conductivity of these samples also at a very low loading of PEDOT:PSS based on percolation theory. This is obvious, otherwise the system would be an insulator. The question is why? Here more morphological and structural characterization of the films should be carried out. Equally important, are these solid solutions? If PEDOT:PSS aggregate, how, what are the domain size? Spectroscopic analysis can help as well IR/Raman.

Response: We thank the reviewer for constructive advice. Although PEDOT:PSS has mechanical flexibility, the PEDOT chains are prone to phase separation and aggregation during processing, due to its rigid conjugated backbone and strong interchain interactions (π - π interaction).

The results of the additional experiments and the the following discussion were also added in the revised supplementary file (Page 3):

“By the introduction of SMS, the interactions between SMS and PEDOT chain (Electrostatic action and hydrogen bonding action) help to reduce PEDOT interchain interactions and to reduce aggregation. At the same concentration of PEDOT:PSS, the good dispersion of PEDOT will improve the uniformity of film and increase conductivity. However, the conductivity of SACPs still follows the percolation theory. To present a further morphological and structural characterization of the SACPs, we carried out DLS, Raman, XRD, and XPS tests. As shown in Supplementary Figure 2c, the particle size distribution of PEDOT:PSS solution was slightly changed after the addition of SMS, indicating that SMS had no negative effects on the dispersion stability of PEDOT. This result was also proved from TEM and SEM images (Supplementary Figure 2d and e). The domain size of PEDOT:PSS was about 40 nm. According to the XRD test, the diffraction peak was left shift from 25 ° to 18° as the addition of SMS, indicating an increase of the corresponding space of PEDOT (Supplementary Figure 2d and e). The aggregation of PEDOT was significantly reduced in SACPs because of the good dispersion stability of PEDOT after the addition of SMS. The elements and interactions were also confirmed by the Raman spectrum (Supplementary Figure 2g) and XPS analysis (S2P) (Supplementary Figure 2h).”

Figure R1-2. Representative chemical and microstructural characterizations of SACP composites: **a**, Size distribution of PEDOT in pure PEDOT:PSS aqueous and SACP solution from the tests of dynamic light scattering; **b-c**, TEM (**b**) and SEM (**c**) images of SACP; **d**, XRD analysis of pure PEDOT:PSS film and SACP film; **e**, Raman spectrum of pure PEDOT:PSS and SACP film with 3.6% mass ratio of PEDOT:PSS; **f**, The binding energy of S_{2p} from XPS spectrum of pure PEDOT:PSS (black line) and SACP film (red line).

3. Ref 44 should be discussed a bit more in the context of the work. Also, close analysis of the plot of Figure 2h seems not reconcile with the data reported in panel a and the conductivity of panel f.

Response: Thanks for the suggestion. We have added further discussion of Ref. 44 in the revised manuscript. Indeed, Ref. 44 developed effective and practical methods to prepare conducting polymer composites with superior mechanical performance (high stretchability and low modulus) via the introduction of ionic liquids into PEDOT:PSS and water-borne polyurethane composites.

Thanks for the reviewer’s careful review of Figure 2h. We are sorry for confusing you in reading these results. The data reported in Figure 2h was coincident with Figure 2a and Figure 2f. To better identify these data in Figure 2h, we provided the number of these results in **Figure R1-3**.

To make it clear, we have added the following text in the revised manuscript in Page 8:

“Recently, to produce elastic PEDOT:PSS composites with low young’s modulus and high conductivity, ionic liquids plasticizers were used to adjust the microstructure of PEDOT complexes. For example, polyurethane and ionic liquid-doped PEDOT: PSS composites have been achieved at a modulus of 7 MPa and electrical conductivity of 140 S/cm⁴⁴.”

Figure R1-3. **a**, Stress-strain curves of the free-standing films with different PEDOT: PSS loadings. **b**, Conductivity of the films with different PEDOT:PSS loadings. **c**, Relationships of Young's modulus and conductivity versus fracture strain of a variety of PEDOT: PSS composites reported in references and this work.

4. The description of the SACP formulation preparation in lines 90-94 is confusing. Also, what is reported in Methods is not clear. Data are for many different samples fabricated in a way that is not reported. The experimental procedure for each samples has to be crystal clear. The same in some of the figures/text when referring to PEDOT:PSS (0.9% to 36.3% mass ration of PEDOT:PSS – versus WHAT?)

Response: Thanks very much for the suggestion. We are sorry for the unclear description of our experimental methods. We have revised the manuscript carefully and added a Supplementary Table about compositions of the SACPs to show experiment details.

1) Page 5: "SACPs consist of three components, *i.e.*, supramolecular solvents (citric acid and cyclodextrin with a molar ratio of 10:1), elastic polymer networks (chemically crosslinked PVA networks with GA), and conductive polymers (PEDOT:PSS) (Figure 1a). Briefly, citric acid, cyclodextrin, PVA, and GA were successively added to the aqueous solution of PEDOT: PSS."

2) Page 16: "The SACP films were prepared by drying SACP precursor solution. The SACP precursor solution was prepared by mixing SMS solution, polymer solution, and conductive polymer solution. SMS Solution comprised of β -Cyclodextrin (β -CD, AR, Aladdin) and Citric Acid (H_3Cit , AR, Aladdin), which were mixed in water at the molar ratio of 1:10 under agitation³⁴. Polymer solution contained poly (vinyl alcohol) (PVA 1799, DP of 1700, 98%-99% hydrolyzed, Aladdin) aqueous solution as polymer network and glutaraldehyde (GA, 50% volume percentage, Macklin) as cross-linker. Finally, SACP precursor solutions were obtained by adding SMS solution and polymer solution into PEDOT: PSS aqueous solution (1.1–1.3% solid content, Clevios™ PH1000, Heraeus Electronic Materials) under vigorous agitation for 10 min. The crosslinking reaction of GA and PVA was triggered in presence of H_3Cit as an accelerator. The crosslinking density of the PVA network was controlled by adjusting the added volume of GA. The details of SACPs with various mass ratios of PEDOT:PSS were presented in Supplementary Table 1."

3) The mass ratio of PEDOT:PSS means that the mass ratio of the mass of PEDOT:PSS versus the mass of SACPs. To make it clear, we have added the following text in the revised manuscript (Page 16):

“The mass ratio of PEDOT:PSS (X) in SACPs was calculated according to the following equation: $X = \frac{X_0 m_0}{X_0 m_0 + m_{SMS} + m_{PVA} + m_{GA}} \times 100\%$; X_0 is the weight percentage of PEDOT:PSS in original solution ($\approx 1.3\%$); m_0 is the weight of PEDOT:PSS solution; $m_{(SMS)}$, $m_{(PVA)}$, and $m_{(GA)}$ is the weight of supramolecular solvent, PVA, and GA, respectively.”

Table R1-1. The details of various SCAPs with different PEDOT:PSS mass ratios.

Mass ratio	SMS solution			Polymer solution		PEDOT:PSS/g
	β -CD/g	Critic acid/g	H ₂ O/g	10% PVA/g	50%GA/ μ L	
0.9%	0.59	1	5	1.2	2	1.25
1.8%	0.59	1	5	1.2	2	2.5
3.6%	0.59	1	5	1.2	2	5
7.1%	0.295	0.5	2.5	0.6	1	5
13.2%	0.295	0.5	2.5	0.6	1	10
23.3%	0.295	0.5	2.5	0.6	1	20
36.3%	0.295	0.5	2.5	0.6	1	37.5

5. Skin is discussed in several places. What is the modulus range needed for the human skin. Please report.

Response: Thanks for the suggestion. As reported in the literature (Ref. 16: Nat Rev Mater 2020, 5, 351-370), the elastic modulus of human skin is about 60-850 kPa.

We have added the modulus range of the human skin in the revised manuscript (Page 3):

“However, biological tissues, *e.g.*, skins and muscles, are typically soft (with a mechanical modulus of 60-850 kPa), and the contact areas interfacing with electrodes are always irregular and even dynamic¹⁶.”

6. Line 138, should be insulating not insulated.

Response: Thanks for pointing out this mistake. We have corrected it.

7. Report viscosity range of formulations in the main text.

Response: Thanks for the suggestion. The viscosity of un-crosslinked SACP precursor solution is 243.6 mPa s, and the corresponding crosslinked SACP precursor is 11883.5 mPa s.

we have added the viscosity value in the revised manuscript (Page 6):

“The components are all soluble in water and capable of forming a homogeneous, stable, and viscosity-tunable aqueous ink (with viscosity on the order of 10^3 to 10^4 mPa s, **Figure S5**),”

8. Experiments were carried out by placing the SACP films in contact with a human subject. This area of research is strongly regulated. One thing if a materials is inserted in a device and then the substrate is placed in contact with the

skin. Here, I have concerns in carrying out experiments on a human without addressing safety concerns. For instance, what is the pH of these films? How long are they in contact with the human subject? Does this person understand the risk of this experiment?

Response: We thank very much the reviewer's comments. We aim at developing biocompatible conductive polymer adhesive. SACPs are comprised of PEDOT:PSS, citric acid, β -cyclodextrin, and crosslinked PVA, which are safe and has been widely applied in the many studies. Thus, SACPs can be considered as safe polymer adhesives. The human subjects in this work are Mr. P. Tan and Dr. H F. Wang, who are the co-first author of this work. Both of the two subjects are informed of the potential risks in the experiment and know about that. The testing time was less than 1 hour in EMG tests and 10 to 20 min in the adhesion demo. The pH value of SACPs precursor solution is 3 to 4 due to the presence of citric acid.

To make it clear, we have added a sentence in the revised manuscript (Page 16):

“The pH value of SACPs precursor solution was about 3 to 4 due to the presence of citric acid.”

9. The method section must be improved. Details for formulation preparation should be included. Also, for the mechanical tests, what is the RH? In general how many samples were measured for reporting statistical data?

Response: Thanks for the suggestions. We have revised the method section carefully accordingly.

To make it clear, we added measurement detail information in the revised manuscript (Page 18):

“The measurement was conducted at 25 °C with a relative humidity of 40%-45%. The reported statistical data were obtained from at least 3 times tests of the same sample.”

Response to Reviewer #2:

Reviewer #2 (Remarks to the Author):

In this manuscript, the authors developed a self-adhesive conductive polymer (SACP) composite based on PEDOT:PSS, PVA, GA, and biocompatible supramolecular solvent. Furthermore, the authors investigated the mechanical and electrical properties of the composite film and demonstrated future practical applications such as electrophysiological sensing electrodes and advanced integrated systems based on the film. Although the concepts of this work, figures, and demonstration in this manuscript are interesting and exquisite, the reviewer cannot find enough strength in terms of scientific novelty and practical specialty. Especially, the highest softness, the highest adhesive, and the highest conductivity of the film show in different composite ratios, which represents that there is a trade-off. In addition, the authors did not even that optimize such properties. Moreover, the reviewer found quite many concerns from this manuscript. Therefore, the reviewer thinks this manuscript is not suitable for a high-ranked journal such as Nature Communications. More specific comments are listed below.

Response: Thanks for the reviewer's constructive comments and kind suggestions. Indeed, they are helpful for us to improve the quality of this work. We have conducted several additional experiments and revised the manuscript accordingly. The point-to-point responses to these comments are presented below.

1. The authors mentioned that “PVA polymer networks are constructed in PEDOT: PSS composites to obtain large yet reversible stretchability.”. However, the reviewer believes that the authors need to describe more experimental results such as morphologies to support the claim.

Response: We thank the reviewer's suggestions. We have added structural and chemical characterizations in the revised manuscript. We conducted an additional dynamic light scattering (DLS) test to measure the changes of PEDOT domain size. By combing the results from TEM and SEM tests, the results indicated that the aggregation of PEDOT was significantly reduced in SACPs because of the good dispersion stability of PEDOT after the addition of SMS. The AFM (Figure S3) and rheology tests (Figure S4) were performed for the characterization of SACPs. Raman spectrum, XRD, and XPS characterizations were also supplied in the revised Supplementary Figure 2c-h.

To make it clear, we have added discussions of structural and chemical characterizations in the revised Supplementary file (Page 3):

“By the introduction of SMS, the interactions between SMS and PEDOT chain (Electrostatic action and hydrogen bonding action) help to reduce PEDOT interchain interactions and to reduce aggregation. At the same concentration of PEDOT:PSS, the good dispersion of PEDOT will improve the uniformity of film and increase conductivity. However, the conductivity of SACPs still follows the percolation theory. To present a further morphological and structural characterization of the SACPs, we carried out the DLS, Raman, XRD, and XPS tests. As shown in Supplementary Figure 2c, the particle size distribution of PEDOT:PSS solution was slightly changed after the addition of SMS, indicating that SMS had no negative effects on the dispersion stability of PEDOT. This result was also proved from TEM and SEM images (Supplementary Figure 2d and e). The domain size of PEDOT:PSS was about 40 nm. According to the XRD test, the diffraction peak was left shift from 25 ° to 18° as the addition of SMS, indicating an increase of the corresponding space of PEDOT (Supplementary Figure 2d and e). The aggregation of PEDOT was significantly reduced in SACPs because of the good dispersion stability of PEDOT after the addition of SMS. The elements and interactions were also confirmed by the Raman spectrum (Supplementary Figure 2g) and XPS analysis (S_{2p}) (Supplementary Figure 2h).”

Figure R2-1. Representative chemical and microstructural characterizations of SACP composites: **a**, Size distribution of PEDOT in pure PEDOT:PSS aqueous and SACPs solution from the tests of dynamic light scattering; **b-c**, TEM (**b**) and SEM (**c**) images of SCAPs; **d**, XRD analysis of pure PEDOT:PSS film and SACPs film; **e**, Raman spectrum of pure PEDOT:PSS and SACPs film with 3.6% mass ratio of PEDOT:PSS; **f**, The binding energy of S_{2p} from XPS spectrum of pure PEDOT:PSS (black line) and SACPs film (red line).

2. In Fig. 2, the author demonstrated the mechanical and electrical performance of SACP film based on only a 3.6 % mass ratio. The reason seems that the mass-ratio-based film shows the highest fracture stress. However, no description to clarify it in the manuscript.

Response: Thanks for the comments. And we are sorry for not clearly explaining our results. In our work, we aim to design a material that can build conformal electrical interfaces on soft surfaces. And thus we prepared SACP as bioelectrode to meet the requirements in electrical and mechanical properties of a conformal electrical interface as discussed in Supplementary Figure 1. To verify the performance changes with various PEDOT:PSS loading in SACP, we showed the mechanical and electrical performance of SACP with various PEDOT:PSS loadings (from 0.9% to 36.6%) in Figure 2a and 2f. As shown in Figure 2a and 2b, at a high mass ratio of PEDOT:PSS, the SACP show high modulus and high fracture stress. Thus the mechanical property was not suitable for conformal electrical interface. As discussed in Figure 3f, SACP show good interface adhesion strength and low young's modulus at a mass ratio less than 7.1%. Although the conductivity of SACP was not the highest at a 3.6% mass ratio, this value is high enough to meet the requirements of bioelectrode (the conductivity of SACP was much greater than that of skins).

To make it clear, we explained the choice of SACP film with a 3.6% mass ratio in the revised manuscript (Page 9):

“Taking into account the compromise in mechanical flexibility, conductivity, and interface adhesion (in following discussions), SACP with PEDOT:PSS mass ratio of 3.6% presented suitable mechanical property (modulus of 401.9 kPa) and conductivity (3.79 s/cm) meet the requirements of bioelectrode.”

3. In addition, it is helpful to understand the film properties comprehensively if the mechanical properties of 36.3% mass-ratio-based film are investigated because the film shows the best conductivity.

Response: We thank the reviewer's suggestion. We agree with the reviewer's opinion. The mechanical performance of SACP was strongly dependent on the mass ratio of PEDOT:PSS. As shown in Figure 2, when the mass ratio of PEDOT:PSS increases, the SACP became hard (modulus and fracture stress increased). The mechanical property of SACP with 36.3% mass ratio of PEDOT:PSS is hard with a fracture strain of 154%, Young's modulus of 5.9 MPa, and fracture stress of 14.8 MPa (Figure 2a and 2b). While, the mechanical property of SACP with a 3.6% mass ratio of PEDOT:PSS is soft with a fracture strain of 736%, young's modulus of 401.9 kPa, and fracture stress of 1.2 MPa. These results indicate that SMS doping provides an effective strategy to soften the PEDOT: PSS composites.

We added discussions in the revised manuscript (Page 6, 7):

“The mechanical property of SACP with 36.3% mass ratio of PEDOT:PSS is hard with fracture strain of 154%, Young's modulus of 5.9 MPa, and fracture stress of 14.8 MPa. While, the mechanical property of SACP with a 3.6% mass ratio of PEDOT:PSS is soft with fracture strain of 736%, Young's modulus of 401.9 kPa, and fracture stress of 1.2 MPa. We observe a significant decrease in Young's modulus and an increase in stretchability after doping. As the mass ratio of PEDOT: PSS decreases from 3.6% to 0.9%, the elastic modulus and fracture stress of SACP gradually decrease to 56.1 ± 13 kPa and 593.2 ± 178.2 kPa.”

4. The SACP film cannot return to its original length after stretching ~400% in Fig.2c. Based on Supplementary Fig.6, it seems that the film was undergone plastic deformation. So, Fig2d needed to be obtained by different samples but no description in the manuscript.

Response: Thanks for the comments. All the samples tested in Figure 2c and 2d are the same SACP films with a 3.6% mass ratio of PEDOT:PSS. As shown in Figure 2c, the SACP film was only 1mm longer (about 5% elongation) after stretching. Shape change in both ends of the SACP film was caused by the finger press, not by stretching. The elasticity can also be proved in Supplemental Video 3, the SACP film adhered on PI film can rapidly recover under

repeated stretching-releasing cycles. According to the stress-strain test, we can only find plastic deformation of SACP film with PEDOT:PSS mass ratio over 7.1%.

To clearly describe this result, we added a description of the sample in the revised manuscript (Page 7):

“SACPs with a 3.6% mass ratio of PEDOT:PSS show a large range of reversible stretchability (400%) (Figure 2c), and the residual strain is less than 50% even at a large tensile strain (Figure 2d and 2e).”

5. Why the thinnest film shows the highest conductivity? Typically, the thicker film has better conductivity, especially such micrometer-thick range. However, the author needs to clarify it to improve the quality of this manuscript.

Response: Thanks very much for pointing out this and we are sorry for not describing the experimental details clearly. Notably, several parameters can affect the performance of SACP films, such as thickness and preparation methods. In our experiment, thin SACP films (4, 7, 11 μm) were prepared by spin-coating, while thick ones (60, 160, 270, and 340 μm) were prepared by casting the solution in a PC mold. SACP films prepared by spin-coating and casting should have some differences, which may lead to some variation of the conductivity. There are several ongoing projects about the application of the as-made SACP thin film as adhesive and transparent electrodes for flexible optical and electronic devices. In these projects, we are currently working on fine-tuning the parameter and conditions for the fabrication of ultrathin SACP films to meet the requirements of several optical and electronic devices. And we will report the progress soon in the future.

To make it clear, we have revised the experimental method part (Page 16):

“For thick SACP films, we applied a casting method. Briefly, after degassing in a vacuum desiccator for 20 min, the SACP dispersion was poured into a polycarbonate (PC) mold and dried at 40 $^{\circ}\text{C}$ for 24 hours. Thus, free-standing SACP films were obtained. The film thicknesses were controlled by changing the volume of the SACP precursor solution. For SACP thin films, we applied a spin-coating method. In a typical experiment,”

We also revised the following discussion in the revised manuscript (Page 8):

“Besides, the as-made composites can be prepared as conductive films. Interestingly, the as-made SACP films with different thicknesses (4 - 340 μm) by spin-coating (for thin-film) and casting (for thick-film) methods showed good uniformity in electrical conductivity (Figure 2g).”

6. Label for the x-axis of Fig. 2h should not be a strain but the fracture tensile strain.

Response: We thank the reviewer's suggestion. We have revised Fig 2h accordingly as shown in Figure R2-2.

Figure R2-2 Relationships of Young's modulus and conductivity versus fracture strain of a variety of PEDOT:PSS composites reported in references and this work.

7. The authors should clearly describe the liver surface of a pig when they demonstrate the applicability of the film adhesion onto the biological tissue since the wet or dry surface of the organ has a huge difference in terms of adhesion.

Response: Thanks for the comment and we are sorry for not describing it clearly. We agree that the adhesion between wet and dry surfaces has differences. In the present study, the live surface was cleaned with absorbent paper and thus the surface is dry. The current SACP is not suitable for adhesion on wet surfaces. We thank the reviewer's suggestion again, we are currently working on varying the structures and compositions of the SACP to meet the requirement of adhesion on wet surfaces as well as wet environments.

To make it clear, we have added this instruction in the revised manuscript (Page 27):

“Image of adhesion between adhesion film and liver of a pig (the live surface was cleaned by absorbent paper)”

8. Fig. 3g shows quite a different adhesion force of the film depending on substrate. However, the authors never mention and try to investigate the scientific reason. Such a lack of chemistry weakens the quality of the manuscript.

Response: Thanks for the constructive comments. We agree with the reviewer's opinion that the mechanism of interface adhesion on various substrates is very important. In this work, we focus on the design of a conformal electrical interface with high interface adhesion strength, low mechanical mismatch, and high conductivity. As shown in figure 3g, the resultant SACP film showed good interface adhesion ability even for different polymer substrates. For the SACP, the interface adhesion force was origin from physical interactions between SACP and substrates

such as electrostatic interaction, hydrogen bond interaction, and Van der Waals interaction, these were also proved in other works (Ref 10: Adv. Mater. (2020) 32, e2001496; Ref. 47: Nat. Commun. (2017) 8, 2218). The differences in interfacial adhesion forces of various polymer substrates were mainly dependent on the molecular polarization strength of polymer chains and molecule structures. Besides, the interface roughness is also an important factor to increase adhesion strength.

We have supplemented the following mechanism discussion in the revised manuscript (Page 10):

“The differences in interfacial adhesion forces of various polymer substrates were mainly dependent on molecular polarization strength of polymer chains and molecule structures^{10, 47}. Besides, the interface roughness is also an important factor affecting adhesion strength.”

9. On page 11, the authors mentioned, “The as-made patterned film exhibits high stability under repeated stretching (Figure 4b)”. However, the authors need to be careful of such description without investigating its conductivity change or microscopic view such as SEM image since it is possible to have lots of microcracks although it is not visible by bare eyes.

Response: Thanks for the reviewer’s comment and suggestion. we have conducted additional experiments to support the as-mentioned description. Previously, we meant that the patterned film had very good adhesion stability even deposited on elastic Ecoflex substrate, which can be proved by images in Figure 4b (bottom) and Supplementary Video 2, and the SEM images also showed no significant changes in surface morphology after stretching. The reviewer’s comment inspired us to further investigate resistance stability. The resistance change was recorded under repeated stretching in **Figure R2-3**, indicating good stability in resistance change.

1) We provided results of resistance change under repeated stretching in the revised supplementary files (Supplementary Figure 9).

2) We added the following discussion on the stability of electrical resistance under dynamic deformation in the revised manuscript (Page 8):

“According to the dynamic electrical resistance variations test (Figure S9), the relative electrical resistance changes less than 1 at a strain of 100%, and the variation values remained constant under repeated strain-releasing cycles. This result indicates that the SACP films present good stability of resistance at a tensile strain within 100%.”

3) We also revised the following text in the revised manuscript (Page 11):

“As shown in Figure 4b (ii) and (iii), the as-made patterned film showed no delamination from Ecoflex substrate under repeated stretching, indicating high adhesion stability of SACPs on the soft dynamic surface.”

Figure R2-3. a, The relative resistance variation ($\Delta R/R_0$) of SACP film under tensile strain. Insert: Magnification

of resistance-strain curve at the strain range of 0-100%. **b-c**, The resistance changes of SACP film under repeated stretching-releasing cycles. Insert: SEM images of the surface before and after the test.

10. The image quality of supplementary Figure 16 needs to be improved. The reviewer cannot get any idea from the figure, although the authors mentioned it is an SACP film.

Response: Thanks for the suggestion. We carried additional experiments the fabrication of large-size SACP thin film as well as its application as transparent yet adhesive thin-film electrodes for ACEL devices. The result is added in Supplemental Figure 16. As shown in **Figure R2-4**, a large-area SACP film (19 cm × 13 cm) was prepared by Meyer rods coating. And this large-area SACP film was applied as a transparent thin-film electrode to assemble a large-area ACEL film ((14 cm × 9 cm). We supplied this result in the revised supplementary files (Supplementary Figure 16).

To make it clear, we added a sentence in the revised manuscript to explain these results (Page 11, 12):

“We also demonstrated the fabrication of large-size (13×19 cm²) transparent yet adhesive thin film on PET substrate by using Meyer rod coating method, and the corresponding SACP’s conductive film can be used as electrodes of a large-area alternating current electroluminescent (ACEL) device with an area of 9×14 cm² (Figure S16).”

Figure R2-4. Image of a large area of self-adhesion SACP film on PET film (a) coated by Meyer rod coating and the corresponding ACEL film (b).

11. To be solid for the author’s description, “The transparent conductive film on the PET substrate was not broken even pasted by commercial test tape several times (Figure 4g).” in page 11, conductivity and microstructure need to be compared before and after repeating peeling tests by commercial tapes.

Response: We thank the reviewer’s suggestion. We followed the suggestion and conducted additional experiments, The results were attached as shown in **Figure R2-5** (Supplementary Figure 17). Indeed, the resistance change was negligible under repeated peeling tests for 6 times, and we didn’t observe obvious microstructural changes in SEM images before and after repeating peeling tests.

1) We revised this sentence in the revised manuscript (Page 12):

“The transparent conductive film on the PET substrate was not broken even pasted by commercial test tape several times and no obvious electrical resistance changes were observed (Figure 4g and Figure S17d).”

2) We also supplied **Figure R2-5** in the revised supplementary files (Supplementary Figure 17).

Figure R2-5. The electrical resistance changes of SACP transparent conductive film under 6 cycles repeating peeling tests. Insert: SEM images of the surface before and after the peeling test.

12. The reviewer suggests a more systematic investigation of the ACEL performance, such as luminance under the different mechanical strain or deformation.

Response: Thanks for the reviewer’s constructive suggestion. We have followed the suggestion and conducted additional experiments to investigate the performance of ACEL devices. We have tested the effects of working voltage and frequency of external electrical bias on the intensity of the ACEL device (**Figure R2-6**). The luminance intensity increased rapidly (**Figure R2-6a**) and the relation between the voltage and emission intensity followed the equation

$L = L_0 \exp(-\beta/V_{-1/2})$, where L is the emission intensity, V is the voltage, and L_0 and β are the constants (Adv Mater (2016) 28, 7200-7203). In addition, the emission intensity increased at a range of frequency of working voltage from 0 to 60 kHz (**Figure R2-6 b**). we also showed the changes in emission intensity under various strains (**Figure R2-7**). The emission intensity increased at low strain while decreasing at high strain because of the trade-off between the resistance and space of the electrode under strain. The electrical field increase as the spacing of electrodes decreases under strain, while the working bias decrease as the resistance increase under strain.

We provided results on ACEL devices characteristics as Supplementary Figure 19 in the revised supplementary file. We also added a discussion in the revised manuscript (Page 13):

“Not limited to the ultra-high flexibility, the ACEL device is highly stretchable when using an elastic substrate, and the emission intensity increased at low strain then decreased at high strain, because the electrical field is varied by the trade-off between the resistance and space of electrode under strain (Figure S19 d-h).”

Figure R2-6. The effect of voltage (a) and frequency (b) of working bias on the emission intensity.

Figure R2-7 . Demonstration of ACEL film under deformations. a-c, Images of folded ACEL film in half; d-f, Images of ACEL film under various stretching states; g, The relationship of emission intensity versus strain of ACEL film; h, Images of ACEL film under the strain of 0%, 100%, 200%, and 300%.

13. In the Methods, the authors need to clearly describe the microfluid molding method and transfer printing through schematic illustration or detailed description.

Response: Thanks for the reviewer's suggestion. we have revised the manuscript according to the suggestion.

In the revised version of the manuscript (Page 17):

“For the microfluid molding method, a PDMS microchannel was obtained by solidifying PDMS precursor (10:1, Sylgard 184, Dow Corning) on a homemade silicon wafer template patterned with a photoresist. Then, the resultant PDMS microchannel was covered by a PET film with a hole punched on the end of the channel. The hole was sealed

with an SACP precursor solution droplet (2-3 g) and then degassed in a vacuum desiccator. Afterward, The SACP precursor solution was injected into PDMS microchannel once the air pressure of the vacuum desiccator gradually returns to atmospheric pressure, due to the pressure difference between the inside and outside of the channel, as detailed in our previous work⁵³. After removing the covered PET film and dried at 40 °C for 24 hours, 3D deposited SACP films were obtained.”.

14. On page 16, the authors have to elucidate what kind of gas-based plasma is used for cleaning since the properties will differ depending on the gases.

Response: Thanks for the suggestion. We are very sorry for not clearly describing this experimental detail.

We have added this detail in the Methods part in the revised manuscript (Page 19):

“The surface treatment was conducted on a plasma cleaning machine (HARRICK Plasma PDC-002) under 400-500 mtorr in air for 15 min.”

15. The reviewer thinks it is better to provide Arduino code for the integrated system in the supplementary information.

Response: Thanks for the suggestion. We have done as suggested in the revised supplementary file. The Arduino code was showed as following:

“Arduino code for the integrated system

```
int ledPin = 10;
int ledPin1 = 8;
int ledPin2 = 7;
int ledPin3 = 6;
int a;
void setup() {
  // put your setup code here, to run once:
  Serial.begin(115200);
  pinMode(ledPin, OUTPUT);
  pinMode(ledPin1, OUTPUT);
  pinMode(ledPin2, OUTPUT);
  pinMode(ledPin3, OUTPUT);
}

void loop() {
  // put your main code here, to run repeatedly:
  a = analogRead(A0);
  delay(200);
  if (a > 350) // a threshold value to define the switches of ACEL array
  { digitalWrite(ledPin, HIGH);
  digitalWrite(ledPin1, HIGH);
  digitalWrite(ledPin2, HIGH);
  digitalWrite(ledPin3, HIGH); // ACEL array controlling
  }
  else if (a > 300)
  { digitalWrite(ledPin1, HIGH);
```

```

digitalWrite(ledPin2, HIGH);
digitalWrite(ledPin3, HIGH);
digitalWrite(ledPin,LOW); // ACEL array controlling
}
else if (a >270)
{ digitalWrite(ledPin2, HIGH);
digitalWrite(ledPin3, HIGH);
digitalWrite(ledPin, LOW);
digitalWrite(ledPin1,LOW);// ACEL array controlling
}
else if(a >230)
{ digitalWrite(ledPin3, HIGH);
digitalWrite(ledPin, LOW);
digitalWrite(ledPin1, LOW);
digitalWrite(ledPin2,LOW);}
else {
    digitalWrite(ledPin, LOW);
    digitalWrite(ledPin1, LOW);
    digitalWrite(ledPin2, LOW);
    digitalWrite(ledPin3, LOW);
}
// ACEL array controlling
}”

```

Response to Reviewer #3:

The work by Tan et al. is exciting and tackles one key challenge that is often overlooked in wearable organic electronics, i.e., the materials surface adhesion. In this manuscript, the authors demonstrated the possibility of obtaining a conducting material capable of acting as tape and withstanding high mechanical stresses. Overall, this manuscript is a perfect fit for Nature Communications and leaves little criticism. There are, however, a few issues that still need to be addressed before publication:

Response: Thanks for the positive comment. The point-to-point responses to these comments are presented below.

1. While the mechanical properties and devices are well characterized, I found the chemical characterization of this polymeric system lacking. The cartoon reported in Figure 1 provides a schematic overview of what could happen in this blend. However, there are no chemical data to support this claim.

Response: Thanks for the constructive suggestions. We have followed the suggestion and carried out several additional experiments about chemical characterization. Details can be referred to the Response of Q1 from Reviewer #2.

2. How stable are these composites in a water and humid environment? The authors briefly mentioned stability in water of the ACEL device, where the top layers provide some packaging effect for the layer underneath. Yet, a proper characterization of the water stability is missing.

Response: Thanks for the suggestion. The raised question is really helpful to improve the quality of this work. The SACPs are water sensitive. At high humidity or doped into the water, the SACPs will absorb water. The presence of water will affect its adhesive capacity. For the ACEL devices, the SACPs electrode is covered with a PDMS layer, which has a packaging effect and can play as a waterproof layer. We found that the ACEL device can still work after soaking in water for at least 24 hours (Figure 4l).

To make it clear, we revised the figure caption of Figure 4 to support this description (Page 28):

“1, Images of the ACEL device immersed in water for 24 hours under bright (i) and dark (ii) backgrounds.”

3. The conductivity values shown here are indeed good. Yet, as the authors correctly reported, there is a trade-off between high electrical conductivity and good mechanical properties. Therefore, a more in-depth analysis of what is more important for the selected applications is needed.

Response: Thanks for the constructive suggestion. The suggestion inspires us to think about the relationship between the properties of the SACPs and their selected applications. In this work, we aim to design a conformal electrical interface suitable for soft and even dynamic surfaces (e.g., adhesive electrodes for EMG monitoring, transparent yet soft electrodes for electronic devices, as well as their integration systems).

For adhesive bio-electrodes for EMG monitoring, first of all, it is required to have good interface adhesion and mechanical flexibility. Second, the conductivity of the electrode should be greater than that of the human body (at the level of mS/cm) and interface impedance is as small as possible. As presented in Figure 2, the SACP has significant improvement on the mechanical flexibility and ductility of PEDOT:PSS. The elastic modulus of the SACPs is less than 1 MPa and the adhesion strength is greater than 100 N/m when the PEDOT:PSS content is less than 7.1%. To meet the mechanical requirements in bioelectrodes, the PEDOT:PSS contents cannot be above 7.1%. at the same time, the conductivity of all these SACPs materials is larger enough to support the application in bioelectrode. For the application as transparent yet adhesive electrodes for soft electronic devices, the adhesion property of the electrode should facilitate the fabrication and conformal packaging of durable soft electronic devices and the solution-process manner provides a potential for large-scale production capability.

We added a discussion in the revised manuscript (Page 9), the following statement is added:

“Taking into account the compromise in mechanical flexibility, conductivity and interface adhesion (in following discussions), SACPs with PEDOT:PSS mass ratio of 3.6% presented suitable mechanical property (modulus of 401.9 kPa) and conductivity (3.79 s/cm) meet the requirements of bioelectrode.”

4) The state of the art is well discussed. Yet, some recent literature on self-healing conducting polymers (see for example, Adv. Funct. Mater. 2019, 29, 1905426 and Adv. Funct. Mater. 2020, 30, 2002853) is missing, and it is unclear how the current work relates to those. What are the self-healing properties of the conducting polymers under investigation?

Response: Thanks for the positive comment. As the reviewer stated, self-healing conducting polymers are very important. In this work, we found that two SACP films can be adhered strongly (Figure 1c and Figure 4h) and can bear a huge shear force, relying on their interface adhesion capability. We found that the self-healing performance of our study may not be an advantage compared to its interface adhesion ability. While this suggestion gives us a new idea that we may try to design an SMS strategy to achieve self-healing conducting polymers in future work.

We have added the references in the revised manuscript (Page 12).

“Similar to the superiority of self-healing organic conducting film on electrical recovery after mechanical damage⁴⁹⁻

⁵⁰, this SACP film may also open a way to electrical recovery.”

Overall, this is an interesting manuscript, and the data reported here deserve publication in Nature Communications.

Response: Thanks again for the positive comment. The comments raised by the reviewer are really helpful for us to improve the quality of this work.

REVIEWER COMMENTS

Reviewer #1 (Remarks to the Author):

The authors have addressed all my technical points in full. I think the safety issues of these tests on individuals remain unclear, however, this does not impact the quality of the work. In my opinion, it could be published in the current form.

Reviewer #2 (Remarks to the Author):

Overall, the authors have provided further explanations for most of my comments, and I think the manuscript has been improved. However, I still have several concerns about the author's responses. I consider this manuscript publishable in Nature Communications if the author address remained concerns sufficiently.

1. In previous my comment #1, I asked about experimental evidence for PVA polymer networks in PEDOT:PSS. But the author responded about the interaction between SMS and PEDOT chain. Although the author revised this manuscript, I still think the author needs to provide an experimental/analytical result, such as FT-IR.

2. Although the author further explained the reason for the different adhesion forces of the film on various substrates (for my comment #8), I believe a more detailed description is needed to clarify it. The author explained too briefly to support the adhesion force trend if Fig.3g.

Reviewer #3 (Remarks to the Author):

The revised manuscript convincingly addresses the reviewer's concerns so that I can recommend publication in Nature Communications.

Response to Reviewer #1:

The authors have addressed all my technical points in full. I think the safety issues of these tests on individuals remain unclear, however, this does not impact the quality of the work. In my opinion, it could be published in the current form.

Response: We would like to thank the reviewer again for the positive comments and recommendation. The previous comments and suggestions help us to improve the manuscript.

Response to Reviewer #2:

Overall, the authors have provided further explanations for most of my comments, and I think the manuscript has been improved. However, I still have several concerns about the author's responses. I consider this manuscript publishable in Nature Communications if the author address remained concerns sufficiently.

Response: Thank you very much for your positive and constructive comments to help us to improve our study.

1. In previous my comment #1, I asked about experimental evidence for PVA polymer networks in PEDOT:PSS. But the author responded about the interaction between SMS and PEDOT chain. Although the author revised this manuscript, I still think the author needs to provide an experimental/analytical result, such as FT-IR.

Response: Thanks for the kind suggestion. And we are sorry for not clearly describing the PVA networks in PEDOT: PSS composites. We have followed the suggestion and conducted additional FT-IR characterization of the PEDOT: PSS, PVA/GA, SMS, and SACPs in the revised manuscript. The results have been added as **Figure S4** in the supplementary file. Indeed, the results further verify the PVA network in SACPs. Besides, the viscosity of SACP precursor had an obvious increase (**Figure S6**), and the stretchability and resilience of resultant SACPs (**Figure S5**) can be improved, as the formation of chemically crosslinked PVA networks. Furthermore, the trend of the elastic modulus of SACPs by increasing the amount of crosslinker (GA) (**Figure S7**).

Supplementary Figure 4 | The FT-IR spectra of PEDOT: PSS, chemically crosslinked PVA with glutaraldehyde (GA) (PVA/GA), supramolecular solvents (SMS) and SACPs. For the PVA/GA, a broad band at 1456 cm^{-1} belongs to the stretching vibration of $-\text{CH}_2-$ of PVA, and a weak peak at 844 cm^{-1} is attributed to the bending vibration of $-\text{CH}_2-$ of PVA. Notably, we can find a specific band at 1089 cm^{-1} , which belongs to the stretching vibration of $-\text{C}-\text{O}-$ of PVA. For SMS, the peak at 1710 cm^{-1} belongs to the $-\text{COOH}$ groups of citric acid, and the bands at 1016 , 1188 and 1402 cm^{-1} are assigned to the stretching vibration of $-\text{C}-\text{OH}$ groups. All the peaks in FT-IR spectra of SMS can be found in that of SACPs, indicating the presence of SMS in SACPs. Importantly, we can find a weak band at 1089 cm^{-1} specific (light green area) to PVA/GA in the SACPs, demonstrating the formation of PVA polymer networks in PEDOT: PSS.

2. Although the author further explained the reason for the different adhesion forces of the film on various substrates (for my comment #8), I believe a more detailed description is needed to clarify it. The author explained too briefly to support the adhesion force trend if Fig.3g.

Response: Thanks for the additional comments. To make it clear, we added the following changes in the revised manuscript (Page 10):

“The differences in interfacial adhesion forces of various polymer substrates were mainly dependent on molecular polarization strength of polymer chains and variations in chemical structures.^{10,47} The high adhesion strength on PI or PEEK substrate can be ascribed to the strong dipole interactions of $-\text{C}=\text{O}$ groups of their polymer primary chains. Meanwhile, PE or PDMS substrate has low interface adhesion strength due to their weak dipole interactions of polymer primary chains. Besides, the interface roughness and interface energy of substrate are also important factors affecting adhesion strength.”

Reviewer #3 (Remarks to the Author):

The revised manuscript convincingly addresses the reviewer's concerns so that I can recommend publication in Nature communications.

Response: We would like to thank the reviewer again for the positive comments, especially the previous comments and suggestions help us to improve the manuscript

REVIEWER COMMENTS

Reviewer #2 (Remarks to the Author):

The author had sufficiently addressed all the reviewer's comments. I believe the manuscript is well revised and improved. I recommend accepting this manuscript for publication in Nature Communications.